# Proteogenomic network analysis reveals dysregulated mechanisms and potential mediators in Parkinson's disease

Abolfazl Doostparast Torshizi ®[1] ✉, Dongnhu T. Truong[1], Liping Hou[1], Bart Smets[2], Christopher D. Whelan[3] & Shuwei Li[1]

Parkinson's disease is highly heterogeneous across disease symptoms, clinical manifestations and progression trajectories, hampering the identification of therapeutic targets. Despite knowledge gleaned from genetics analysis, dysregulated proteome mechanisms stemming from genetic aberrations remain underexplored. In this study, we develop a three-phase system-level proteogenomic analytical framework to characterize disease-associated proteins and dysregulated mechanisms. Proteogenomic analysis identified 577 proteins that enrich for Parkinson's disease-related pathways, such as cytokine receptor interactions and lysosomal function. Converging lines of evidence identified nine proteins, including LGALS3, CSNK2A1, SMPD3, STX4, APOA2, PAFAH1B3, LDLR, HSPB1, BRK1, with potential roles in disease pathogenesis. This study leverages the largest population-scale proteomics dataset, the UK Biobank Pharma Proteomics Project, to characterize genetically-driven protein disturbances associated with Parkinson's disease. Taken together, our work contributes to better understanding of genome-proteome dynamics in Parkinson's disease and sets a paradigm to identify potential indirect mediators connected to GWAS signals for complex neurodegenerative disorders.

Parkinson's disease (PD) is a neurodegenerative disorder affecting ~1 million individuals in the US[1] with growing global burdens projected to affect 12.9 million individuals by 2040[2]. As the second most common neurodegenerative disorder[3], PD onset is directly associated with age, peaking at ages 85 to 89 years with higher prevalence in men[4]. Clinically, PD patients suffer from motor and non-motor indications causing severe complications in their daily lives. Pathologically, striatal dopamine deficiency caused by neuronal loss in substantia nigra as well as the presence of Lewy bodies in the midbrain are the hallmarks of PD[5,6].

Although advancing age is the greatest risk factor for PD[2], genetic and environmental factors also play a critical role in disease risk and progression. For instance, despite complexity of characterizing environmental factors, pesticides, exposure to heavy metals, smoking and caffeine have been reported to be linked to the PD risk[7–9]. On the other hand, genetic factors for PD identified in the past decade range from rare mutations underpinning familial disease[10–12] to large-scale genome-wide association studies (GWAS) unmasking the polygenic influences on idiopathic PD[1,13]. While common genetic variants account for almost 22% of the genetic heritability of PD, the genetic loci identified to date only explain a limited fraction of this estimate[2]. Addressing this gap requires an up to three-fold increase in the number of cases[1]. Importantly, elucidating the biological foundations and functional impacts of PD risk variants requires investigating their interconnections with their corresponding protein products. This is critical given that although human genetics is an important approach for target identification/validation in pharmaceutical research, it remains imprecise[14] as GWAS signals frequently fail to implicate the causal

[1]Population Analytics & Insights, AI/ML, Data Science & Digital Health, Janssen Research & Development, LLC, Spring House, PA, USA. [2]Neuroscience Data Science, Janssen Pharmaceutica NV, Beerse, Belgium. [3]Neuroscience Data Science, Janssen Research & Development, LLC, Cambridge, MA, USA. ✉ e-mail: adoostpa@its.jnj.com

genes mediating the disease[15,16] or map to putative drug targets whose underlying biology is poorly understood[17]. Establishing functional impacts of PD risk variants at the proteomic level may help inform/prioritize therapeutic targets and potential biomarkers of PD.

Proteins underpin most of the regulatory processes and act as the structural pillars of cells[18]. Protein functions are not determined solely by their abundance but by several types of properties, including trafficking between distinct microdomains, interactions with other proteins, or undergoing posttranslational modifications[19]. Moreover, proteomics can partially capture the impact of environmental toxins on brain function[20]. Therefore, they have great potential as biomarkers, therapeutic targets, and surrogates for disease progression as well as a means for endotype identification[21]. Until recently, high-throughput proteomic profiling of large cohorts has been challenging due to technical limitations. Recent advances in population-scale proteomic studies such as The UK Biobank Pharma Proteomics Project (UKB-PPP)[14] have facilitated larger-scale investigations of genome-proteome interactions enabling more thorough investigations for target discovery efforts.

Given the polygenic etiology and clinical heterogeneity of PD, systems-level analyses of proteomic data jointly with genetics and other omics data modalities may unmask the major driving factors and interacting molecules that contribute to the disease. In this study, we have leveraged large-scale genomic data from the UK Biobank (UKB)[22] and FinnGen[23] as well as population-scale proteomics data from UKB-PPP to conduct a systems-level proteogenomic analysis to investigate interrelations of genome and proteome as well as identifying proteins whose genetically-driven disruptions can contribute to PD.

## Results

### Study design

Identifying causal genes that can be further considered as potential therapeutic targets remains a complex undertaking[24]. Characterizing causal variants that underpin genetic associations with a disease, as well as elucidating the mechanistic impacts of genetic aberrations on the human proteome and normal protein interactions necessitates integrative approaches to combine genetics with other omics data types such as proteomics and epigenomics[25,26]. Moreover, since majority of drug targets are proteins, then proteome-based Mendelian randomization (MR) as well as modeling alterations in the human interactome caused by genetic abnormalities is of great importance in terms of identifying novel disease causal genes/proteins[27,28]. In this

study, we designed a three-phase framework (Fig. 1) to characterize the impact of PD genetic variants on proteins followed by a series of network-based analyses, epigenomic and transcriptomic analyses aimed at creating a comprehensive mapping of interconnections between genome and proteome and systemically integrate such data-types to identify proteins that might be associated with PD. Such a systems-level design using orthogonal data from the same cohort of patients allowed us to detect not only direct genetic risk factors, but potential indirect mediators for which no sentinel genetic variants were identified that could play critical roles in disease etiology and pathology. Such mediators may either be proteins identified through causal inference analysis or through network analysis. The designed workflow consists of (1) proteogenomic analysis; (2) network analysis; (3) transcriptome/epigenome data analysis. The outcome of the proteogenomic analysis is subsequently utilized in the following phases.

The proteogenomic analysis consists of six steps as follows: (1) genome-wide association analysis (GWAS) in UKB on PD; (2) meta analyzing the summary statistics of the UKB GWAS with the GWAS results from FinnGen; (3) protein quantitative trait loci (pQTL) mapping across 54,219 individuals and 2923 proteins in UKB; (4) causal inference; (5) characterizing the enriched biological pathways by the identified proteins followed by determining the pathway specificity of the PD risk variants; (6) aggregating the identified risk variants and generating gene-based PD association scores to be used in network analysis.

Using the gene-based scores from the proteogenomic analysis phase, we leveraged a publicly available protein-protein interaction network (PPI)[29] to run a series of network analyzes including: (1) a hypothesis-free approach, where we mapped the gene-based scores onto the PPI, conducted an exhaustive search to identify PD-associated risk modules, and paired the most significant modules with pQTLs to pinpoint dysregulated proteins; (2) a hypothesis-driven approach, where we mapped the genetic and proteomic signals to identify pQTL-associated proteins interacting with the genetics-derived risk proteins; (3) characterizing the enriched biological pathways by the identified PD-associated protein complexes. Finally, in the third phase, we conducted epigenomic and single-cell transcriptomic analyzes to investigate the regulatory roles of the identified genetic signals as well as determining the cell-specific transcriptional patterns of PD risk factors in the human brain.

In the following sections, we will describe the details of each step of this study design along with the findings.

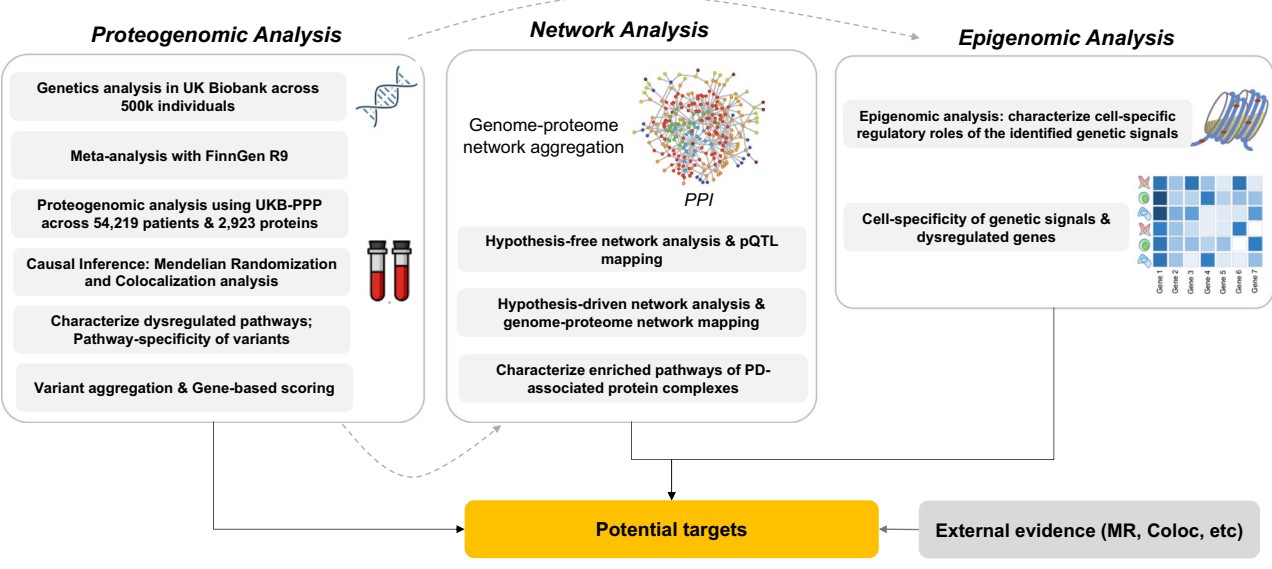

**Fig. 1 | An overview of the study design.** The study consists of three stages, including proteogenomic analysis, network analysis, and epigenomic data analysis.

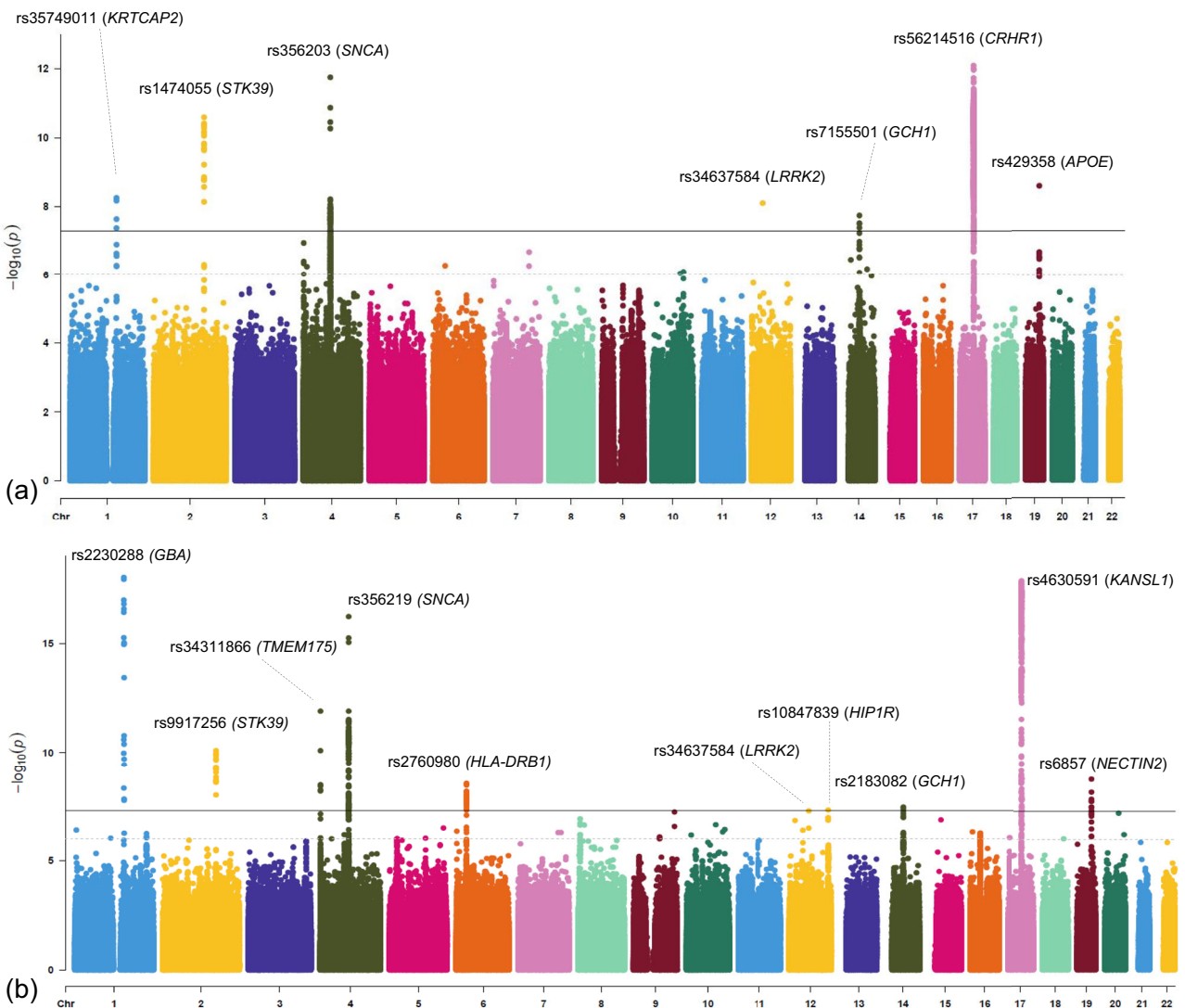

**Fig. 2 | Manhattan plots for PD case-control GWAS using the EHR-based clinically defined cohorts from the UKB and the UKB-FinnGen meta-analysis in European samples.** (**a**) UKB case-control GWAS, (**b**) UKB-FinnGen GWAS meta-analysis. A solid blue line indicates the significance threshold P < 5×10⁻⁸. The dashed line shows P = 10⁻⁶.

## Proteogenomic analysis

Using UKB clinical and imputed array data, we created a cohort of 2,864 clinically defined PD patients and 158,876 controls of European (EUR) ancestry and conducted GWAS using REGENIE v2.2.4[30] (Methods, Supplementary Data 1). A list of top identified significant associations are provided in Fig. 2a and Table 1. Details of the conducted GWAS are provided in the Supplementary Methods.

We next conducted a meta-analysis (Methods, Supplementary Figs.) of the UKB-curated cohort and FinnGen R9 (Table 2, Fig. 2b). FinnGen R9 contains 4,235 PD cases and 373,042 healthy controls leading to a total 7,099 PD cases and 531,918 healthy controls in this study. Overall, 11,114,080 SNPs shared between the two datasets were evaluated for meta-analysis. Details of the performed meta-analysis is provided in the Supplementary Methods. Meta-analyzing UKB and FinnGen has added significant power to replicating many of known PD signals. The FinnGen GWAS signals itself shows two significant loci *UBQLN4* and *ARL17B* in chromosomes 1 and 17, respectively. Upon meta analyzing it with the UKB GWAS, four other significant loci reached genome-wide significance including *HIP1R*, *HLA-DRB1*, *GBA*, and *TMEM175*. These established loci have been identified in the literature mainly in EUR population. Therefore, meta-analyzing UKB with FinnGen not only did not lead to spurious associations, but increased the number of loci reaching genome-wide significance. This can be

attributed to similarities in the genetic architecture of the samples from the UKB and participants of FinnGen. We should note that all the samples in the UKB GWAS were selected from British ancestry.

To investigate if any of the identified significant loci are supported by other studies, we used the largest available meta-analysis GWAS by Nalls et al.[1]. Focusing on the top significant variants in UKB GWAS and the meta-analysis GWAS, seven variants have been replicated, including (a) UKB GWAS: rs1474055-T (intergenic to *STK39*), rs356203-T (intronic to *SNCA*), rs35749011-A (intergenic to *KRTCAP2*), and rs34637584-A (missense to *LRRK2*); (b) UKB/FinnGen meta-analysis: rs4630591-T (intronic to *KANSL1*), rs34311866-C (missense to *TMEM175*), and rs34637584-A (missense to *LRRK2*). Given the low allele frequency of rs34637584 in the imputed array data (MAF EUR < 0.001), we also checked Gene Bass[31] and verified findings through analysis of whole exome sequencing data (OR = 3.37, P = 3.81E-8). The 95% credible set is provided in Supplementary Data 1.

We acknowledge that the conducted meta-analysis sample size is modest compared to existing studies. However, our objective was to align genetic associations using orthogonal data sets generated from the same samples and proteome. Moreover, in order to generate gene-based scores to be used in the network analysis, meta-analysis was performed so that we later used the generated scores to weight the nodes of the PPIs. This became possible by leveraging high-throughput

**Table 1 | Genome-wide significant loci from GWAS performed on the PD cohorts in UKB EUR[*]**

| SNP-effect allele | Closest Gene | MAF[a] | Beta | P | Effect | V2G[b] | CADD[c] | RDB[d] | ClinVar[e] |
|---|---|---|---|---|---|---|---|---|---|
| rs35749011-A | KRTCAP2 | 0.015 | 0.55 | 2.3E-8 | intergenic | 0.15 | 0.64 | 5 | NA |
| rs34637584-A | LRRK2 | 0.002 | 2.27 | 7.9E-9 | missense | 0.3 | 28.2 | 3a | Pathogenic |
| rs1474055-T | STK39 | 0.12 | 0.26 | 1.6E-10 | intergenic | 0.25 | 1 | 7 | NA |
| rs356203-T | SNCA | 0.63 | −0.18 | 5.4E-11 | intronic | 0.21 | 6.6 | 7 | NA |
| rs7155501-G | GCH1 | 0.44 | −0.15 | 1.8E-8 | intronic | 0.16 | 4.4 | 4 | NA |
| rs56214516-C | CRHR1 | 0.2 | −0.26 | 8E-13 | intronic | 0.23 | 5.7 | 5 | NA |
| rs429358-C | APOE | 0.14 | 0.22 | 2.5E-9 | missense | 0.4 | 16.65 | 4 | Pathogenic/likely pathogenic |

[a]*MAF*: Minimum allele frequency.
[b]V2G score extracted from OpenTargets[98], which represents a quantile rank for a given SNP, relative to all other SNPs, mapping to a gene based on the following evidence: in silico functional prediction from VEP, eQTL, pQTL, PC Hi-C, Enhancer-TSS correlation, DHS-promoter correlation, Canonical TSS.
[c]Combined Annotation Dependent Depletion (CADD[99]) score >10 corresponds to the top 10% most deleterious alterations in the genome; CADD > 20 to the top 1% and CADD > 30 to the top 0.1%.
[d]RegulomeDB (RDB[100]) scores:1a-1f = likely to affect binding and linked to expression of a gene target; 2a–2c = likely to affect binding; 3a–3b = less likely to affect binding; 4–7 = minimal binding evidence.
[e]Landrum et al.[101].
[*]The top SNP in each locus is reported.

**Table 2 | Genome-wide significant loci from the meta-analysis of the UKB and FinnGen PD cohorts[a]**

| SNP-effect allele | Closest Gene | MAF-FinnGen | Beta | P | Effect | FM-PP[b] | V2G | CADD | RDB | ClinVar |
|---|---|---|---|---|---|---|---|---|---|---|
| rs2230288-T | GBA | 0.04 | 0.4 | 9.7E-19 | missense | 0.9 | 0.26 | 16.14 | 1 f | NA |
| rs9917256-A | STK39 | 0.16 | 0.15 | 8E-11 | intergenic | 0.85 | 0.19 | 0.37 | 7 | NA |
| rs34311866-C | TMEM175 | 0.2 | 0.17 | 1.25E-12 | missense | 1 | 0.36 | 12.9 | 4 | NA |
| rs1372519-G | SNCA | 0.8 | 0.18 | 3E-12 | 5′ UTR variant | 0.9 | 0.36 | 4 | 4 | NA |
| rs2760980-A | HLA-DRB1 | 0.15 | −0.15 | 2.8E-9 | intergenic | 0.6 | NA | 7.2 | 7 | NA |
| rs34637584-A | LRRK2 | 0.002 | 2.27 | 7.16E-12 | missense | 0.9 | 0.3 | 28.2 | 3a | Pathogenic |
| rs10847839-C | HIP1R | 0.34 | −0.14 | 4.7E-8 | intronic | 1 | 0.11 | 1.6 | 2b | NA |
| rs11158026-T | GCH1 | 0.4 | −0.1 | 3.4E-8 | intronic | 0.95 | 0.21 | 1.2 | 1d | NA |
| rs4630591-T | KANSL1 | 0.8 | −0.25 | 1.4E-18 | intronic | 0.9 | 0.45 | 1.1 | 1 f | NA |
| rs6857-T | NECTIN2 | 0.19 | 0.13 | 1.7E-9 | 3′ UTR variant | 0.75 | 0.17 | 2 | 5 | NA |

[a]The top SNP in each locus is reported.
[b]FM-PP: Fine-mapping posterior probability.

proteomic data from the UKB-PPP. Within our created case/control cohort used for running GWAS in the UKB, 638 PD patients and 15,522 healthy controls had proteomic data. In order to integrate PD genetic association with protein interactome for downstream candidate target identification, we used MAGMA[32] to generate per-gene scores based on the GWAS meta-analysis results (Supplementary Data 2, Methods). We used the SNPs in the body of each gene to calculate a gene-based p-value associated with PD, which was then used in the network analysis.

Using results generated by the UKB-PPP consortium, we evaluated if any of the Parkinson's disease susceptibility variants also act as pQTLs in blood plasma (Methods). Briefly, the UKB-PPP results comprise 14,287 primary genetic associations derived from 2,923 plasma proteins measured across 54,219 individuals[14]. Details of proteomic data generation and downstream quality control are outlined in [14]. We intersected all GWAS significant variants from our meta-analysis of UKB and FinnGen with all UKB-PPP pQTL results. In total, 103,118 associations between genetic signals and protein abundances were found, translating to a total of 577 proteins (Supplementary Data 1) where 10% of the observed associations were *cis* (Fig. 3a). These proteins may not necessarily be disease-associated, thus we further conducted causal inference analyses in the MR and colocalization section to nominate putative causal proteins.

Focusing on the GWAS meta-analysis top signals (Table 2), seven PD-associated SNPs were associated with a total of 163 proteins, including rs2230288, rs34311866, rs2760980, rs10847839, rs2183082, rs4630591, and rs6857, among which 4 SNPs were only associated with one protein including rs2230288-T (Ephrin, EFNA; P = 2.4E-10), rs34311866-C (Alpha-L-Iduronidase, IDUA; P = 4.6E-8), rs10847839

(Huntingtin-interacting protein 1-related protein, HIP1R; P = 6.7E-30), and rs2183082-T (Galectin-3, LGALS3; -log10(P) = 311.6). Three other SNPS including rs2760980-A, rs4630591-T, and rs6857-T were associated with 95, 21, 50 proteins, respectively (Fig. 3b). All pQTLs associating with a single protein were in *cis*.

We sought to investigate the enrichment(s) of those proteins across distinct biological pathways. Accordingly, we conducted pathway enrichment analysis (Methods) on 577 proteins associated with GWAS signals and focused on the top 10 significantly enriched pathway at false discovery rate (FDR) of less than 0.05 (Supplementary Fig. 1). We found several PD-related mechanisms to be enriched in the identified proteins, including lysosome (FDR = 1.6E-6), cytokine-cytokine receptor interaction (FDR = 2.2E-16), and chemokine signaling pathway (FDR = 1.2E-6). Details of the enriched pathways are provided in Supplementary Data 3. Among the observed pathways, recent studies have been increasingly focusing on the role of lysosomal genes as risk factors for idiopathic PD[33,34]. We identified 21 proteins enriched for lysosome pathway including cathepsins (CTSC, CTSD, CTSE, CTSF, CTSL, CTSO, CTSV, CTSZ), ARSA, GLB, GUSB, IDUA, IGF2R, LAMP3, NAGA, NAGPA, PLA2G15, PSAPL1, SCARB2, SGSH, and SMPD1. In addition to known PD-related pathways, majority of the other significant pathways were enriched for inflammatory markers including chemokines such as chemokine ligand (CCL) and CXCL family as well as cytokines such as tumor necrosis factor (TNF) and interleukin (IL). 'Rheumatoid arthritis'-related pathways were also predominantly enriched for inflammatory markers. This is in line with the existing knowledge of multiple shared genetic variants between PD and rheumatoid arthritis and several other autoimmune diseases[35,36]. We point

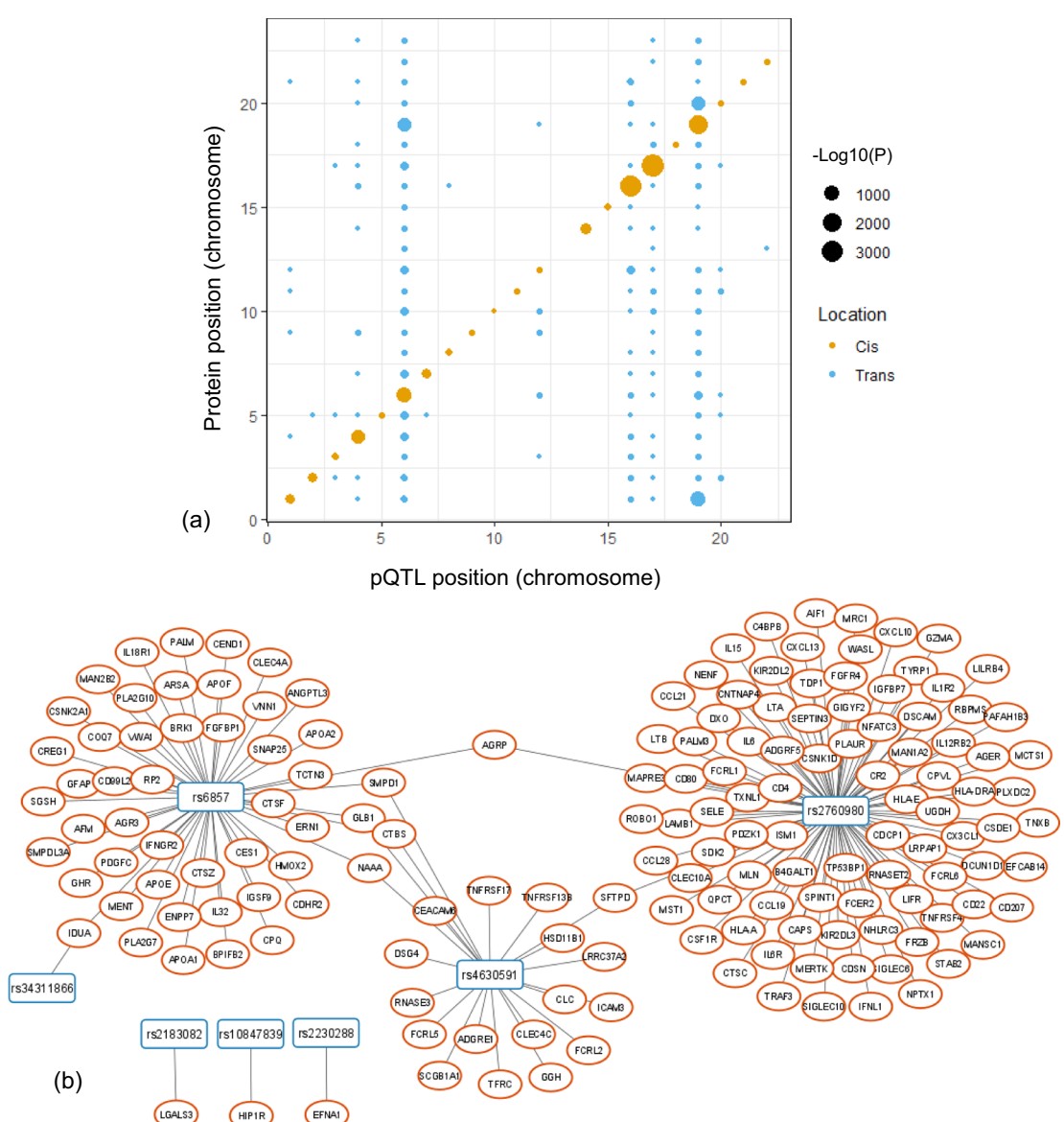

**Fig. 3 | Chromosomal positions of the identified pQTLs relative to their associated proteins. a** Position of pQTLs and their associated proteins; **b** PD-associated SNPs and their associated proteins.

out that the UKB-PPP is a population-based study with proteomic evaluation based on a commercial panel of 2923 proteins, therefore many known PD-related proteins were not profiled in the consortium. The proteins in the Olink panel include low-abundant inflammatory proteins, proteins actively secreted into blood circulation, approved and ongoing drug targets, organ-specific proteins leaked into blood circulation, and proteins representing exploratory biomarkers. As a result, some known PD-related pathways may not show significant enrichment as the relevant proteins were not assayed. Moreover, we conducted an enrichment analysis on the SNPs that were found to be pQTLs in the UKB-PPP using MAGMA[32] to account for potential confounding factors caused by linkage disequilibrium (LD). We observed a few enriched pathways including cytokine-cytokine receptor interaction (FDR = 1.7E-5), chemokine signaling pathway (FDR = 2.7E-5), lysosome (FDR = 1.5E-7), and rheumatoid arthritis (FDR = 2.3E-5). These observations are in accordance with our pathway enrichment results applied to the pQTL-associated proteins.

The UKB-FinnGen meta-analysis is restricted in terms of the sample size. Therefore, we sought to leverage the summary statistics provided by Nalls et al.[1] and evaluate their potential associations with plasma protein concentrations in the UKB-PPP. Among the 107 SNPs provided by Nalls et al., 51 SNPs were pQTLs in the UKB-PPP associated with a total of 204 proteins from 287 pQTL-protein associations (Supplementary Data 1). Out of 204 identified proteins, 149 proteins were shared with the 577 PD pQTL-associated proteins from UKB-FinnGen, while 55 proteins were not identified based on UKB-FinnGen GWAS. pQTL-associated proteins are the proteins whose abundance is significantly associated with the SNPs identified in the UKB-FinnGen meta-analysis. We conducted a pathway enrichment analysis on the 204 proteins leading to a total of 6 significant pathways (Supplementary Data 3). Among these, three pathways were shared with the enriched pathways from the PD associated proteins from UKB-FinnGen, including lysosome (FDR = 4.62E-4), graft-versus-host disease (FDR = 0.003), and cell adhesion molecules (CAMs) (FDR = 0.03). The other three pathways that are unique to this analysis include antigen processing and presentation (FDR = 2.88E-5), staphylococcus aureus infection (FDR = 3.94E-4), and natural killer cell mediated cytotoxicity (FDR = 0.03).

We conducted a Mendelian Randomization analysis to identify potential PD causally-linked proteins. We used 6381 identified cis-

**Table 3 | The identified plasma proteins putatively causal for PD with significant MR and colocalization evidences**

| Gene Symbol | Method | Beta | SE | FDR | PP.H4 |
|---|---|---|---|---|---|
| *PYDC1* | Wald ratio | 1.228 | 0.2006 | 8.5E-09 | 1 |
| *PRSS8* | Wald ratio | −2.719 | 0.4443 | 8.8E-09 | 0.99 |
| *CTF1* | Wald ratio | 2.092 | 0.3418 | 1.11E-08 | 0.99 |
| *CLC* | Wald ratio | 2.16 | 0.3755 | 6.29E-08 | 0.99 |
| *MB* | Wald ratio | 1.671 | 0.3784 | 4.4E-05 | 0.99 |
| *BPIFA2* | Wald ratio | 1.833 | 0.4146 | 4.7E-5 | 0.98 |
| *CD1C* | Inverse variance weighted | 1.062 | 0.2703 | 0.000304 | 0.97 |
| *HLA-E* | Inverse variance weighted | 0.4452 | 0.1136 | 0.00031 | 0.97 |
| *GRN* | Wald ratio | −0.4759 | 0.1272 | 0.000589 | 0.95 |
| *TNFSF13* | Wald ratio | 0.8259 | 0.2375 | 0.00138 | 0.95 |
| *TNFRSF17* | Wald ratio | 1.598 | 0.4623 | 0.0015 | 0.95 |
| *SPINK8* | Wald ratio | −0.4102 | 0.1488 | 0.0125 | 0.95 |
| *KIR2DL3* | Weighted median | 0.1929 | 0.08313 | 0.0411 | 0.95 |
| *CD207* | Wald ratio | −1.373 | 0.4108 | 0.002 | 0.95 |
| *LTB* | Weighted median | 0.3819 | 0.1601 | 3.5E-02 | 0.95 |
| *LTB* | Inverse variance weighted | 0.4924 | 0.2179 | 4.74E-02 | 0.95 |
| *BAG3* | Wald ratio | 1.491 | 0.3375 | 4.19E-05 | 0.94 |
| *CXCL10* | Wald ratio | 2.207 | 0.4003 | 2.03E-07 | 0.94 |

pQTLs from the UKB-PPP as the instrumental variables and protein abundance as the Exposure in this analysis (Supplementary Data 1). We took advantage of summary statistics from Nalls et al.[1] to use in the MR analysis as the outcome (Methods). We identified a total of 122 proteins showing a significant association at FDR < 0.05. We also tested for horizontal pleiotropy on the MR results, where no significant pleiotropic effect was observed. Following MR, we conducted a colocalization analysis between the UKB-FinnGen meta-GWAS and pQTLs from the UKB-PPP to characterize if meta-analysis signals colocalize with pQTL hits in the MR-derived proteins aimed at assessing whether the genetic factors from meta-GWAS also colocalize with the genetic predictor of the protein abundances for supporting the MR findings. We found strong colocalization support (H4.PP > 0.9) for a shared causal variant for abundance and PD risk for 17 proteins identified through MR. These 17 proteins are shown in Table 3 and the full list of all MR results as well as sensitivity analysis results are provided in Supplementary Data 1. In addition, we ran a colocalization analysis on the UKB-FinnGen meta-analysis genes (Table 2) and Nalls et al. signals. Among which *HIP1R* (H4 posterior probability, H4.PP = 1), *GCH1* (H4.PP = 0.99), *SNCA* (H4.PP = 1), *TMEM175* (H4.PP = 0.95) show a significant colocalization signal. We performed another colocalization analysis between the pQTLs from the UKB-PPP and GWAS results by Nalls et al.[1]. We found three loci *BST1* (PP.H4 = 0.95), *GPNMB* (PP.H4 = 0.94) and *CTSB* (PP.H4 = 0.96) with showing a colocalization support (H4.PP > 0.9). These three genes were also found as significant MR evidences in our analysis. Integrating the MR results with the colocalization signals from the UKB-FinnGen meta-analysis and Nalls et al. signals as well as the pQTLs from the UKB-PPP and Nalls et al., we observe four proteins including GPNMB, BST1, CTSB, and HIP1R to be shared. Among which, GPNMB and BST1 have also been reported in CSF as significant MR signals by Kaiser et al.[21]. Given that our findings are based on plasma, such an overlap shows the great potential of these two proteins to be investigated as therapeutic targets.

To gain a better insight into the SNP-protein distribution, we performed a specificity analysis of the PD-associated SNPs identified as pQTLs for the proteins profiled in UKB. Initially, we included all genome-wide significant variants from the UKB-FinnGen meta-analysis

GWAS, then we narrowed down the analysis to the top significant hits reported in Table 2 (n = 10). Among all the significant variants, the number of proteins per pQTL ranged from 1 to 125 while the number of pQTLs per protein ranged from 1 to 2913 (Figs. 4a, b). The frequencies of *cis* associations (defined as ±1 kb flanking regions of gene boundaries) as well as *trans* associations per pQTL were significantly different, where the number of former associations per pQTL ranged from 1 to 16. On the other hand, *trans* associations ranged from 1 to 109 and 798 pQTLs were only associated in *trans* (Fig. 4d). Next, we checked the frequency of variants per proteins in both *cis* as *trans* associations. We found that 540 proteins had only *trans* associations and 37 proteins had both types of associations (Fig. 4c).

We investigated the specificity of the SNPs reported in Table 2 to proteins profiled in the UKB-PPP. rs2230288 (missense variant in *GBA*) shows one significant association in *cis* with EFNA1. rs34311866 (missense variant in *TMEM175*) was observed to be significantly associate with IDUA in *cis* (Beta = -0.05, P = 4.6E-8). Of note, rs2760980 (an intergenic variant close to *HLA-DRB1*) shows 98 significant associations among which five associations were in *cis* while the rest were in *trans*. The *cis*-associated proteins included AGER, AIF1, DXO, HLA-DRA, and TNXB. We found that rs10847839 (an intronic variant in *HIP1R*) is in fact associated with HIP1R in the UKB-PPP (Beta=0.09, P = 6.7E-30). rs11158026 (intronic to *GCH1*) was significantly associated with decreased levels of LGALS3 (Beta = -0.28, -Log10(P) = 316). rs4630591 (intronic to *KANSL1*) shows 21 significant association among which only one association is in *cis* with LRRC37A2 (Beta=1.07, -Log10(P) = 2751) while the remaining association show -Log10(P) < 13. Finally, rs6857 (*NECTIN2*) was found to be highly pleiotropic with 50 significant associations among which only one association in *cis* with APOE was identified (Beta = -0.85, -Log10(P) = 1573). We can see that these variants are mostly specific to significant *cis* associations and a few of them are highly pleiotropic with larger p-values in *trans* associations compared to *cis* associations.

Circulating plasma proteins to study neurological diseases is constrained by the absence of proteins that do not cross the blood-brain barrier (BBB). One alternative tissue for neurological studies is Cerebrospinal Fluid (CSF). Therefore, a systematic comparison between plasma-derived protein measurements versus CSF can potentially shed light on proteins that cannot be captured in plasma either because they cannot make it through the BBB or can make it through at lower concentrations leading to their dilution to noise when preprocessing the data. Leveraging pQTL associations derived from CSF of Parkinson's disease patients by Kaiser et al.[21], we made a systematic evaluation using our identified *cis*-pQTLs and their associated proteins. Please note that there are systematic differences between the proteomic data used in our study as opposed to the data used by Kaiser et al.[21]. UKB-PPP proteomics data is generated by the Olink Explore 3072 antibody-based proximity extension assay (PEA), measuring 2,941 protein analytes capturing 2,923 unique proteins. On the other hand, the data used by Kaiser et al.[21]. is generated by SomaScan aptamer-based proteomics platform[37] which captures 4,135 proteins. The two platforms share a total of 1,583 unique proteins. Please note that due to technical differences of data processing on the two platforms as well as the protocols used for characterizing pQTLs, not all the pQTLs might have been shared between the two studies.

The CSF-based study reports significant *cis* pQTLs for 856 Off-rate Modified Aptamers (SOMAmers) corresponding to 744 unique proteins[21]. Among these proteins, 474 proteins are also profiled in the UKB-PPP data. Comparing the CSF-derived pQTL-associated proteins with our *cis*-pQTL associated proteins, 151 proteins are shared between the two (Supplementary Data 1) leading to a total of 323 proteins that are found to show significant *cis*-pQTL associations only in CSF. We conducted a pathway enrichment analysis on significant proteins unique to CSF using KEGG gene sets[38]. Only one

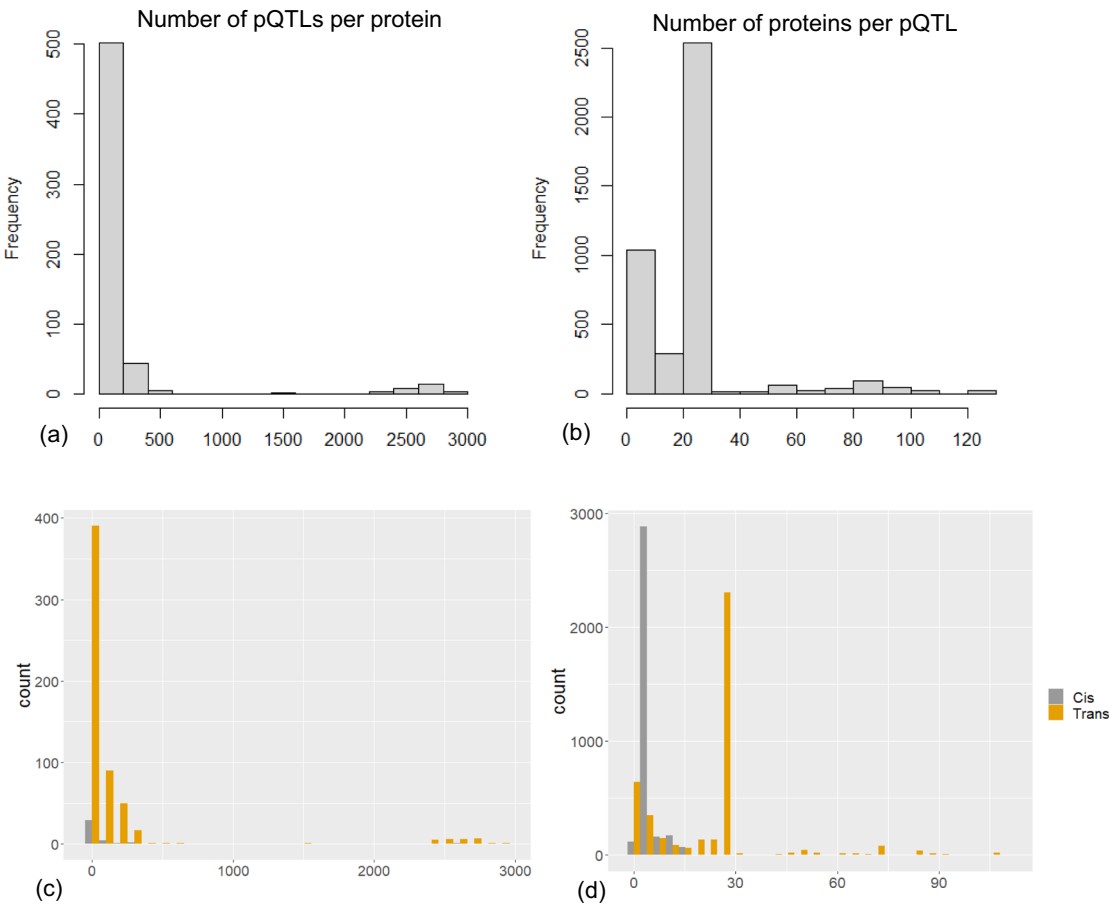

**Fig. 4 | Overview of blood plasma pQTL architecture in PD. a** histogram of the number of PD-associated pQTLs per protein; **b** histogram of the number of associated proteins per pQTL; **c**–**d** number of associations in panels **a**, **b** separated by the association type.

pathway 'Complement and coagulation cascades' (FDR = 0.002) passed the significance threshold of FDR < 0.05. We had not identified this pathway to be significant in our analysis based on the pQTL-associated proteins in plasma. The complement system is a tightly regulated innate immune system playing a key role in regulating the normal function and development of the central nervous system (CNS)[39]. According to Fatoba et al.[39]., accumulating evidence from human postmortem studies demonstrate that neuronal and glia cells are capable of expressing a majority of the complement molecules in CNS. This pathway has been reported in several CNS-related studies such as multiple system atrophy[40], brain proteome profiling of PD in humans and mice[41–43], and psychosis[44].

In addition to evaluating the pQTL-associated proteins in both datasets, we also compared the cis-pQTLs that were identified in both studies. Two pQTLs were shared on both studies including rs429358 and rs557011. In CSF, the missense variant rs429358 was significantly associated with three proteins APOC2 (Beta=0.89, P = 3.56E-39), APOE (Beta=0.82, P = 2.49E-45) and CEACAM19 (Beta=0.46, P = 4.05E-17). However, this variant was only significantly associated with APOE in plasma with a negative direction of effect (Beta = -1.01, -log10(P) = 2069.74). rs557011 was associated with HLA-DQA2 in CSF (Beta=0.53, P = 1.72E-42) while being significantly associated with HLA-DRA in plasma (Beta=0.37, -log10(P) = 455.6). The main difference observed is the opposite association of rs429358 with APOE in plasma versus CSF.

Our observation indicates the intrinsic limitations of measuring circulating plasma proteins to study brain diseases and need for leveraging CNS-specific tissues to shed light on genome-proteome aberrations in PD that may not be captured in plasma.

## Network modeling reveals PD-associated protein complexes

The complex etiology of PD and its polygenic nature requires employing a systems-level approach for joint modeling of distinct data modalities to identify the interactions between the genetic variants with encoded proteins as well as protein-protein interactions that might be disrupted as the outcome of genetic aberrations. Network modeling is among the most efficient tools for modeling complex interactions and analyzing their overall contribution to the disease onset and progression[19,45]. Using the gene-based scores from GWAS meta-analysis of UKB-FinnGen, the identified proteins, and existing protein-protein interaction (PPI) networks, we performed hypothesis-free and hypothesis-driven network analyses to identify protein complexes associated with PD and to characterize potential mediators that indirectly contribute to the disease. In the following, we will discuss the details of each experiment.

To conduct a hypothesis-free network analysis with the goal of identifying potential targets and mediators through the integration of PD-associated genetic signals and PPI networks, we used the GWAS meta-analysis results of UKB-FinnGen and calculated a gene-based score based on the p-values of all the SNPs localizing within the gene boundaries using MAGMA[32]. We employed the PPI network from PICKLE[29], which includes 217,963 interactions containing 16,420 proteins. Then, we used gene-based scores to assign weights to each of the corresponding nodes in a PPI network (Methods). Using the generated weighted PPI network, we identified PD-associated modules using dmGWAS[46]. Here, a 'module' is defined as a subgraph of the larger PPI network with a local maximum proportion of low p-value genes. Upon completing the analysis, each module will have a score based on the gene-based p-values of its member nodes. Overall, 13,897 modules

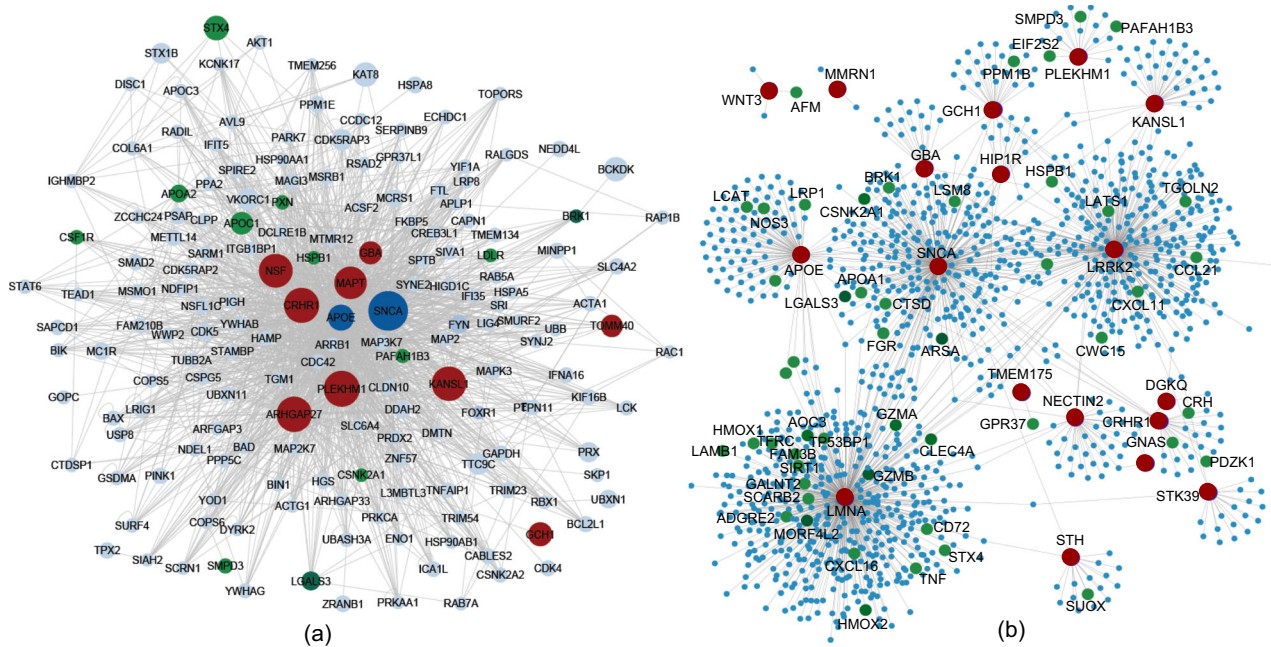

**Fig. 5 | Network analysis results. a** hypothesis-free analysis: aggregated view of the top 1% of the identified PD-associated protein modules. Red nodes represent GWAS signals, green nodes represent pQTL-associated proteins in PD, dark blue nodes are pQTL-associated proteins which are PD-associated GWAS loci, and the size of the nodes is proportional to the gene-based scores generated by MAGMA; **b** hypothesis-driven analysis: aggregated view of the identified loci from UKB-FinnGen meta-analysis GWAS (shown in red) and their first-degree neighbors in the PPI network. Green nodes represent pQTL-associated proteins in PD.

with a varying number of nodes were identified. We sorted the modules based on their scores, focusing on the top 1% significantly PD-associated modules. In total, the top-identified modules contained 175 proteins (Supplementary Data 4). Next, we compared the list of 577 proteins, that were associated with the identified pQTLs, with the identified the proteins in the top 1% of modules. We found 14 proteins (P = 0.003) to be shared between the two groups, including: SNCA, LGALS3, CSF1R, STX4, APOC1, APOE, APOA2, SMPD3, PXN, PAFAH1B3, LDLR, HSPB1, BRK1, and CSNK2A1 (Fig. 5a). Please note that the red nodes shown in Fig. 5a include all the genome-wide significant loci irrespective of being the top locus in the meta-analysis GWAS. Notably, two established familial PD risk loci appeared among the top disease modules only through this PPI analysis – PINK1 and PARK7.

Genetic correlations between PD, Alzheimer's disease, and Lewy body dementia (LBD) have been investigated in the literature[47,48] and a few loci have been reported to be shared between these diseases such as *TMEM175, MAPT, KANSL1*. We sought to identify potential overlaps between the 175 proteins in the top 1% module of the hypothesis-free network analysis and risk genes identified in Alzheimer's Disease. Using GWAS Catalog[49], we found 23 overlapping genes, including *APOE, MAPT, PLEKHM1, KANSL1, NSF, KAT8, BCKDK, APOC1, TOMM40, LRIG1, ICA1L, USP8, UBXN11, FAM210B, AVL9, DISC1, PRKAA1, BIN1, TEAD1, LDLR, CSNK2A1, L3MBTL3,* and *RAC1*. Moreover, we investigated shared genes among the top identified modules with LBD[50] where we observed a significant overlap at P = 1.8E-63. These findings can be further explored for examining the shared molecular signatures among neurodegenerative disorders.

We next conducted a hypothesis-driven network analysis, mapping the UKB-Finngen PD meta-analysis GWAS significant loci onto the human PPI network, and then extracting their first-degree neighbors in the network (Fig. 5b). Among the first-degree neighbors, pQTL-associated proteins identified through proteogenomic analysis are shown in green. In total, 53 proteins were found to interact with PD risk nodes. We ran a set of permutation tests to check if the overlap between the pQTL-associated proteins and the direct neighbors of PD-associated risk loci was not by chance. We randomly selected proteins

from the entire PPI network with a size equal to the network presented in Fig. 5b and tested their overlap with the pQTL-associated proteins for 10,000 iterations. In each iteration, we tested the overlap using Fisher's exact test and created an empirical distribution of the p-values. An empirical p-value of 0.014 was observed, denoting significant enrichment of pQTL-associated proteins within the direct neighbors of PD GWAS loci in the human PPI network. The extracted network in this section was then compared with the top 1% modules in the hypothesis-free network analysis. Considering all the nodes in both networks, we observed a significant overlap between the two (Fisher's exact test P = 7.6E-56). Such observation indicates the importance of considering not only the PD-associated loci from GWAS, but also their interacting proteins whose dysregulation may contribute to the pathways that contribute to the disease. One of the limitations of this analysis is the constrained number of proteins profiled in the UKB-PPP. Therefore, hypothesis-driven analysis may not reflect the full spectrum of the human proteome and as a result, this might have led to overlooking some parts of the human interactome that might be involved in the disease.

**Transcriptomic and Epigenomic data analysis**

In the third phase of our analytical pipeline, we conducted epigenomic and transcriptomic analyses of genes identified from the first two phases. We first studied how GWAS-significant PD risk variants potentially disrupt the expression of linked genes through physical chromatin loops. Second, we investigated how the expression of the identified GWAS loci, as well as their associated proteins in the brain, may be disrupted in PD.

We investigated how the identified PD variants impact disease risk via potential influences on cell type-specific enhancer-promoter interactions using PLAC-seq data by Nott et al.[51]. First, we mapped the identified GWAS SNPs (Fig. 2b) onto the developed cell type-specific interactome, examining whether any variants localized to regulatory regions of their corresponding genes or impacted the regulatory regions of distal genes through physical interactions with chromatin loops.

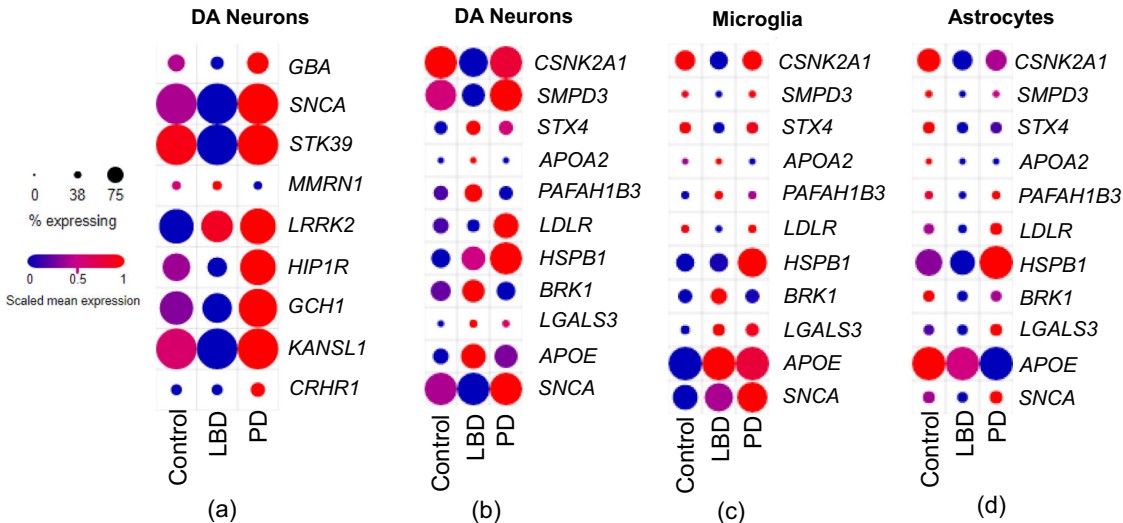

**Fig. 6 | Expression profiles of PD-associated genes among PD, LBD, and healthy individuals. a** Expression of PD-associated genes identified through UKB GWAS and UKB-FinnGen meta-analysis GWAS; **b–d** Expression of the gene encoding the identified proteins from the network analyzes in dopamine (DA) neurons, microglia, and astrocytes, respectively. Source data are provided as a Source Data file.

We found 6 PD variants to be linked with distal genes in cell types essential for central nervous system function (Supplementary Data 5). Two variants were found to exert regulatory effects only in neuronal cells, including rs356219 (intronic variant in *SNCA*) and rs113100008 (intronic variant in *CRHR1*). One variant, rs7542186 (intronic to *LMNA*), was found to act only in oligodendrocytes. Two variants impacted both neuronal cells and oligodendrocytes, including rs10847839 (intronic variant in *HIP1R*) and rs8067056 (intronic to *STH*). Finally, only rs429358 (one of the two *APOE* e4-defining missense variants) was found to be active in microglia. We checked the localization patterns of the linked genes to the PD variants in each cell type. For all the variants, the linked genes localized in the same region. Since the proteomic data in the UKB-PPP is plasma-derived, we sought to investigate if any of the identified cell-specific variants in Supplementary Data 5 localize in open chromatin regions in blood-derived data, to investigate if these variants are brain-specific. We used the assay for transposase-accessible chromatin using sequencing (ATAC-seq) data[52] processed by ATACdb[53] generated from healthy individuals. None of the brain-specific variants were found to be localizing in open chromatin regions in blood implicating that such variants have cell-specific regulatory roles in brain.

To complement the analysis, we conducted a fine-mapping (Methods) on the UKB-FinnGen meta-analysis results aimed at identifying the putative causal variants. Then we integrated them with the data by Nott et al. We consider a variant causal if the fine-mapping posterior probability (PP) > 0.9. We found two variants rs356219 (*SNCA*) and rs10847839 (*HIP1R*) (Supplementary Data 5) to be causal at PP = 1 which were both identified to be acting in neuronal cells. These variants have also been reported by Schilder and Raj[54] to be causal in PD. The remaining variants reported in Supplementary Data 5 did not have a PP > 0.9. Moreover, no other significant variants with PP > 0.9 were found to fall in the chromatin interaction loops provided by Nott et al.[51].

We examined the cell-specificity of the identified PD risk variants as well as their dysregulations in PD patients compared to normal controls. First, we leveraged single-nucleus droplet-based sequencing data from Lake et al.[55] to evaluate if the identified PD risk genes are markers of specific cell-types in distinct brain regions. These data cover >60,000 single nuclei from the human adult visual cortex, frontal cortex, and cerebellum. Using the summary statistics provided by Lake et al.[55], we extracted significantly upregulated genes for each cluster of cells and used them as the marker for that cell type. Among all the PD GWAS risk genes, we found three genes to act as specific markers of certain cell types, including *SNCA* and *LRRK2* in excitatory neurons and *KANSL1* in Purkinje cells.

To further investigate cell specificity in more disease-relevant brain tissues, we leveraged another set of single-cell RNA-seq (scRNA-seq) data from Kamath et al.[56]. These data[56] contain >22,000 dopamine neurons extracted from substantia nigra pars compacta, where ten populations of dopamine neurons were identified. We evaluated how the identified PD loci in Tables 1 and 2 were differentially expressed (DE) between PD, LBD, and healthy control individuals (Fig. 6a). Among the investigated genes, 8 genes were DE in PD compared to healthy controls. Moreover, we observed different dysregulation of several genes in PD versus LBD including *SNCA*, *STK39*, *HIP1R*, *GCH1*, *KANSL1*, which were overexpressed in >75% of cells in PD. However, three other genes *GBA*, *MMRN1*, and *CRHR1* were expressed in less than 50% of cells. Next, we focused on the shared proteins between the hypothesis-free and hypothesis-driven network analyses results and examined how their encoding genes are expressed in PD, LBD, and healthy control individuals in dopamine neurons, microglia, and astrocytes (Figs. 6b–d). These proteins included: SNCA, APOE, CSNK2A1, SMPD3, STX4, APOA2, PAFAH1B3, LDLR, HSPB1, BRK1, and LGALS3. Among the genes encoding these proteins, *HSPB1* was the only gene to be upregulated in >75% of the cells in dopamine neurons, microglia, and astrocytes. *APOE* was upregulated in >75% of cells in microglia within PD and LBD cohorts while it was downregulated in astrocytes in the PD cohort. *SNCA* was significantly upregulated in dopamine neurons and microglia in >75% of cells while it showed upregulation only in ~38% of cells in astrocytes. *SMPD3* was significantly upregulated in dopamine neurons in >75% of cells while showing the same expression patterns in microglia and astrocytes at just a small fraction of cells. *STX4*, *APOA2*, and *PAFAH1B3* showed a stable number of expressing cells across the three cell types with a varying degree of expression levels. *BRK1* shows a larger number of expressing cells in dopamine neurons while less expressing cells in microglia and astrocytes. We found that *BRK1* is significantly downregulated in PD patients compared to controls while being significantly upregulated in LBD versus controls. Finally, we found *LGALS3* to be significantly upregulated in PD and LBD in ~40% of cells in microglia while showing downregulation in LBD in astrocytes.

## Converging lines of evidence suggests 9 proteins as potential mediators of PD

We sought to identify 'indirect' mediators of PD risk–i.e., proteins that may not harbor a PD susceptibility risk variant but may contribute to disease-relevant mechanisms such as neuroinflammation. Based on the proteogenomic analysis and different network modeling results, we defined three sets of targets as follows: (a) pQTL-associated proteins in blood plasma derived from UKB-PPP; (b) proteins that were identified among the top 1% of PD associated modules from the hypothesis-free network analysis whose abundance was also associated with PD pQTLs; (c) PD pQTL-associated proteins which directly interact with the PD meta-analysis GWAS loci in the hypothesis-driven network analysis. Then, we mined the literature to collect additional independent causal inference evidence on each protein. Overall, 9 proteins with converging lines of evidence from the three target sets were identified, including: LGALS3, CSNK2A1, SMPD3, STX4, APOA2, PAFAH1B3, LDLR, HSPB1, BRK1. We screened these proteins for previous peer-reviewed studies linking them to PD and/or other neurodegenerative diseases.

We identified a number of studies reporting links between PD and LDLR[57,58] and CSNK2A1[59]. Two prior studies identified associations between PD and STX4. In the first, Tewhey et al.[60] employed multiplexed reporter assays to investigate the downstream effects of GWAS risk variants across several traits including PD, reporting an association between the PD risk variant, rs11865038, and expression of *STX4*. In the second study, *STX4* was identified as a common risk gene for AD and PD through a transcriptome-wide association study (TWAS)[61]. We further identified three prior studies on the impact of SMPD3 on neurodegeneration and cognitive impairments[62–64]. Several lines of evidence were found about LGALS3. For the remaining nominated proteins, we did not observe published studies to demonstrate their links with neurodegenerative disorders.

We found a series of research studies highlighting the role of LGALS3 in mediating microglial-mediated neuroinflammation in AD[65,66], cognitive function, and neurodegeneration[67–69], and PD[70,71]. Two recent causal inference studies, employing Mendelian randomization (MR), have reported a significant causal association between LGALS3 and PD[21,72]. In the next section, we further explore the potential impact of LGALS3 on the development of PD.

## Potential role of galectin-3 (LGALS3) in PD

As an evolutionarily conserved protein, LGALS3 drives inflammation[73]. The role of LGALS3 in the immune system has been extensively investigated, but its role in the central nervous system is less explored. LGALS3 is observed to be upregulated in several central nervous system diseases associated with inflammation, including AD, hypoxia, and stroke[65,74]. Such an observation is in line with our finding in analyzing the PD scRNA-seq data (Fig. 6). LGALS3 activates microglia and inflammation in diseases such as stroke[75], Huntington's disease[76], and multiple sclerosis (MS) and it is found that it can induce microglial activation only when coupled with tissue damage such as infection or neurodegeneration[74]. Our initial hypothesis about LGALS3 was based on the PD protective variant rs11158026 (beta = −0.1) identified through the meta-analysis of UKB-FinnGen GWAS. Given that rs11158026 was a pQTL associated with downregulation of LGALS3, we hypothesized that this variant is exerting its effect through downregulation of LGALS3. So far, LGALS3 has been explored as a target by a number of teams and companies for diseases such as heart disease, fibrosis, and AD where a few drug compounds have made it to the phase II and III clinical trials[77–79]. However, to our knowledge, no clinical efforts are in place for targeting LGALS3 in brain or inflammatory diseases. Therefore, we believe further investigations to study how inhibition of LGALS3 can ameliorate the footprints of PD is warranted.

LGALS3 is a member of family of galectins which are a set of proteins that share a carbohydrate recognition domain motif which interacts with β-galactoside glycan[80]. Galectins play a role in a wide array of cellular functions including signaling, inflammation, autophagy, and immune response[80]. We queried the GWAS significant SNPs in the proteogenomic analyzes to identify if LGALS3 abundance is associated with any the identified PD-associated variants. We had identified rs11158026-T as a significant SNP with a protective effect (beta = −0.1, P = 3.4E-8) on PD in UKB-FinnGen meta-analysis GWAS (Table 2). We hypothesized that the identified protective association can potentially be explained through the results of proteogenomic analysis and found that rs11158026-T in *GCH1* is a pQTL associated with reduced LGALS3 expression (beta = -0.286, -log10(P) = 316). Therefore, we hypothesized that rs11158026 exerts protective effects against PD by downregulating LGALS3 (Supplementary Fig. 2). A recent study by Burbidge et al.[70] identified that LGALS3 mediates unconventional secretion of SNCA/alpha-synuclein in response to lysosomal membrane damage in human midbrain dopamine neurons. Based on this finding, protective association of rs11158026-T with PD, and its association with reduced expression of LGALS3, we can hypothesize that rs11158026 has a protective role against PD through downregulation of LGALS3. Through MR analysis performed in the Proteogenomic analysis section, we found LGALS3 to significantly associate with PD (beta=0.22, FDR = 0.001) indicating that its overexpression is putatively causal to increased risk of PD. In our MR analysis, rs11158026 was the instrumental variable used in the Wald Ratio test. Testing for potential pleiotropic associations of this SNP, we found no significant pleiotropic association for this variant (Horizontal pleiotropy P-value = 0.94). To further validate this hypothesis, we used two MR analysis results generated from human brain and CSF. In the first analysis, Storm et al.[72] performed an MR using eQTL data from human brain tissue to identify genes with potential causal associations with PD. They identified *LGALS3* (OR = 2.11, FDR = 0.004) as a potential causal gene whose overexpression is associated with increased risk of PD. In a second independent study, Kaiser et al.[21] performed an MR analysis using the existing GWAS summary statistics on PD along with pQTL data from human CSF. They have also identified a causal association between *LGALS3* and PD (beta=0.07, FDR = 7.1E-4), indicating that overexpression of *LGALS3* is significantly associated with increased risk of PD. In fact, the direction of effect between our findings and observations and the developed hypothesis, match both independent MR studies. Convergence of all these independent lines of evidence had led us to nominate LGALS3 as a potential mediator of PD which can indirectly contribute to the PD pathogenesis.

We investigated the abundance of LGALS3 in the UKB-PPP data and analyzed its genotype-specific abundance among PD patients. To gain a deeper insight into the genotype-specific abundance of LGALS3 in pre-diagnosis versus diagnosed PD cases, we divided the PD cohort into 'prevalent' and 'incident' cases. Prevalent cases are patients who have been diagnosed with PD when donating blood samples, while incident cases are individuals who have not been diagnosed with PD when donating blood and were diagnosed later, sometimes years later. Three sets of tests were performed. Initially, we used all the incident and prevalent cases (Fig. 7a) and ran an ANCOVA (analysis of covariance) to evaluate the abundance of LGALS3 between patients with different genotypes at rs11158026. Then, we ran a similar experiment on prevalent and incident cases only (Figs. 7b, c). The combined set of cohorts shows a significant decrease in LGALS3 abundance in homozygous carriers of rs11158026-T compared to the heterozygous carriers. Both were also showing a lower abundance compared to non-carrier patients. Focusing on prevalent cases, we found a significant downregulation of LGALS3 in TT carriers versus non-carriers as well as a marginally significant downregulation in CT compared to CC. However, no significant difference between homozygous and heterozygous carriers was observed (Fig. 7b). Finally, considering only incident cases, we found a significant downregulation in the whole three sets of comparisons (Fig. 7c). Moreover, the number of non-

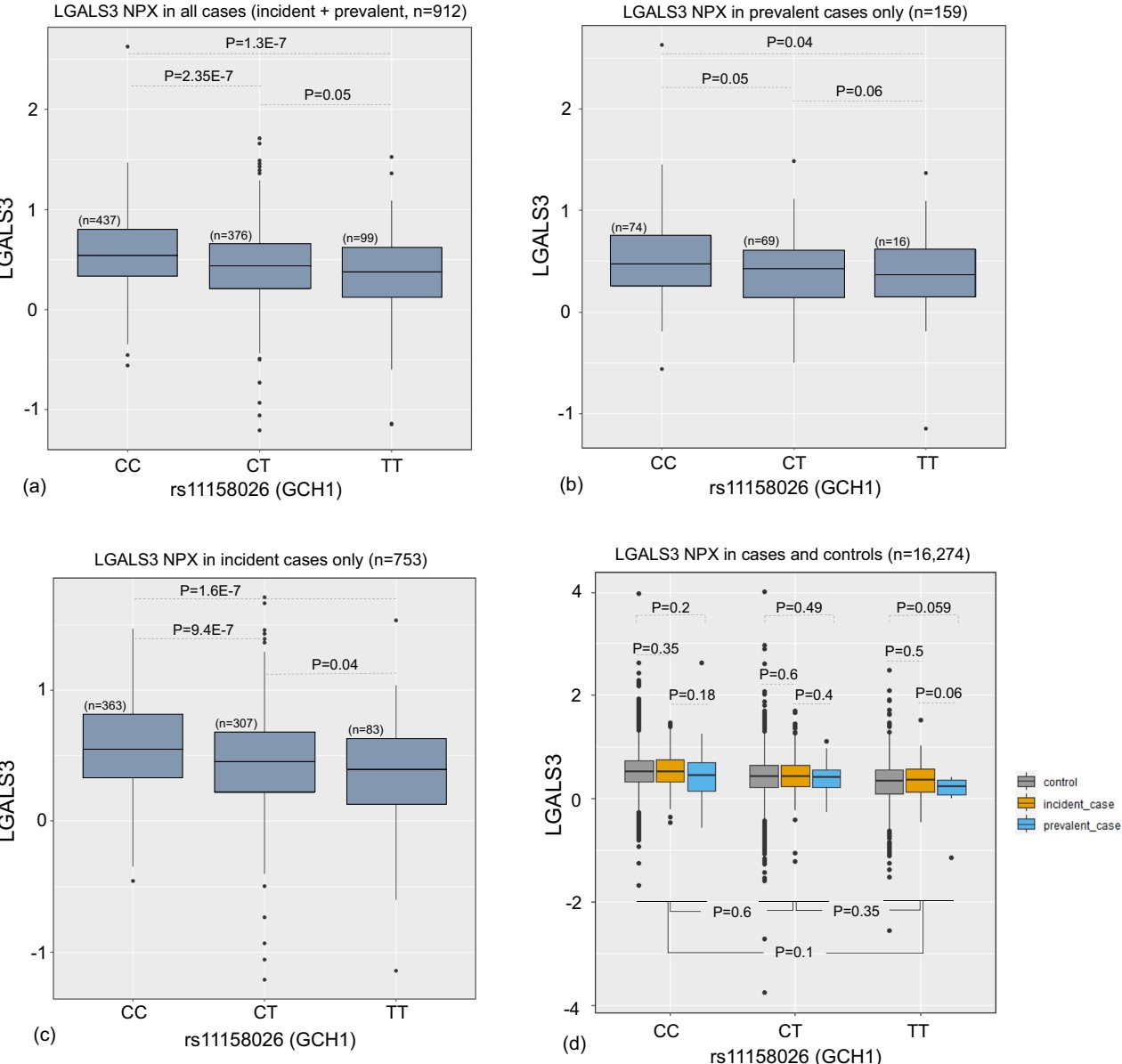

**Fig. 7 | Genotype-specific abundance of LGALS3 among carriers of the protective SNP rs11158026 among. a** incident+prevalent PD cases; **b** prevalent PD cases only; **c** incident PD cases only; **d** cases and controls. NPX: Normalized Protein Expression. In (**d**) the *p*-values indicated at the bottom of the figure represent the differences between the NPX values once combining incident, prevalent and controls in each genotype. All data panels are represented as median values plus the first and third quartiles. Two-sided ANCOVA test was performed to evaluate the genotype-specific abundance of LGALS3 adjusted for age and sex. Source data are provided as a Source Data file.

carriers (*n* = 363) is larger than the carriers among incident cases (CT, *n* = 307; TT, *n* = 83). So we compared the number of carriers and non-carriers between the incident cases and the control cohort in the UK Biobank. We found heterozygous carriers of this variant had a marginally significantly lower proportion of PD incident cases compared to non-carriers (Fisher's exact test *P* = 0.04, OR = 0.8). Similarly, homozygous carriers who later developed PD were significantly less than non-carriers (Fisher's exact test *P* = 0.02, OR = 0.75). Genotype-specific abundance of LGALS3 among the incident cases, prevalent cases, and control individuals were conducted (Fig. 7d). In each genotype, we did not observe a significant difference between the three sets of individuals.

## Discussion

PD is the second most common neurodegenerative disease with extreme health and financial burdens. GWAS has played a crucial role in disentangling the genetic underpinnings of the disease, leading to the discovery of multiple genetic variants associated with overall susceptibility and age of onset of disease. The functional impacts of PD-associated genetic variants at the protein level are relatively under-explored. To gain a deeper insight into the genome-proteome system and its implications in causing the disease, we developed a systems-level proteogenomic analytical pipeline to leverage population-scale genome and proteome data from the UKB and FinnGen aimed at identifying potential direct and indirect therapeutic targets/mediators. In this study, we developed a three-phase pipeline with the following components: (1) proteogenomic analysis; (2) network analysis; (3) epigenomic/transcriptomic data analysis. In phase 1, we conducted a series of genetic association tests using carefully curated cohorts from the UKB and FinnGen, followed by an evaluation of the identified susceptibility SNPs in a large-scale pQTL database constructed by the UKB-PPP consortium. We replicated many established PD GWAS

signals, including *LRRK2, GCH1, GBA, SNCA, TMEM175, STK39, KANSL1*. All the significant PD variants were then subjected to proteomic analysis to identify the proteins whose abundance are significantly associated with the identified PD SNPs. Our analyses led to the identification of 577 proteins enriched for PD-related biological pathways including cytokine-cytokine receptor interactions[81] and lysosomal function[82,83]. Next, we conducted a MR and a colocalization analysis and identified 122 proteins that are putatively causal in PD.

In the second phase, we leveraged gene-based scores from the proteogenomic analysis along with the existing PPI networks to conduct a series of network modeling analyzes aimed at identifying PD-associated network modules in the human protein interactome and integrating the proteogenomic signals with the PPI networks for nominating potential PD targets/mediators. Focusing on the top identified modules in the hypothesis-free network analysis identified 175 proteins, among which 14 proteins were also associated with the identified PD genetic variants. Hypothesis-driven network analysis revealed 53 pQTL-associated proteins to directly interact with established PD GWAS loci, demonstrating downstream effects of susceptibility variants on protein functionality. Finally, in phase 3, we identified 6 PD risk variants to exert regulatory effects on distal genes in specific cell types, including neuronal cells, oligodendrocytes, and microglia. These findings revealed how PD risk variants could have regulatory roles on distal genes, potentially attributable to differences between chromatin-accessible regions in different cell types. For instance, we found that the missense variant rs429358 in *APOE* exerted a regulatory role on several flanking genes in microglia only. This is in line with in-vitro findings on the role of *APOE* in impairing microglial activation and function[84,85]. Epigenomic analysis using blood-derived ATAC-seq data showed that none of the identified variants exert regulatory roles in certain cell types in the brain localized in open chromatic regions, illustrating the brain-specific roles of PD variants. Moreover, we observed how PD-associated genes are dysregulated in certain cell types via scRNA-seq. An important aspect of this analysis was to find the difference between the dysregulation patterns of PD risk genes in PD and LBD in dopamine neurons. For example, in dopamine neurons, we found that *SNCA* is significantly upregulated in PD while showing an opposite direction in LBD, signifying disparities between the two diseases.

A major goal of our study was to identify proteins with a causal link to disease through the integration of genetic, proteomic, and protein interaction data. This allowed us to generate converging lines of evidence for genes that are not directly significant GWAS loci, can contribute to the disease onset or progression. We eventually identified 9 proteins that exhibited converging lines of evidence. Several of these proteins, as well as their encoding genes, have been shown to link with neurodegeneration. Among these findings, several converging lines of evidence have indicated that LGALS3 may be causally associated with PD. Beyond PD, recent studies show that *Apoe* controls the microglial transition from a healthy cell to a toxic neurodegenerative cell through regulating *Lgals3* suggesting that inhibition of *Lgals3* is a neuroprotective strategy for treating glaucoma[86,87]– highlighting it as a potential therapeutic target.

Our study is an unprecedented effort to integrate UKB proteogenomic data in a network framework to characterize the genome-proteome relations and model their interactions. This approach can provide a cohesive perspective on genetically driven protein disruptions contributing towards PD and other complex diseases. A major limitation is the sample size of our PD susceptibility GWAS. Additionally, while blood is an accessible sample source enabling population-scale proteogenomics, proteomic data generated from CSF and/or brain tissue may be more informative for target discovery in neurodegeneration. Our analysis is also limited to the 2923 unique protein analytes from the Olink panel; until proteomic coverage increases, the network modeling employed in this study may allow us to look beyond

the assayed proteins. Moreover, analyzing the interactions between genetic risk variants with environmental factors followed by characterizing their aggregate impact on the human proteome are among the future research goals of this study. An expansion avenue for the future steps will be to use published large-scale GWAS studies (of other diseases) to test against the protein level in the UKB-PPP. This work lays the groundwork to gain a bigger picture about the reflection of genetic aberrations on the human proteome and how such affected proteins can be further studied and validated for future purposes such as target discovery and biomarker identification.

## Methods
### Statistics & reproducibility
The UK Biobank is a population-based, longitudinal, biomedical cohort consisting of over 500,000 individuals that link demographic, clinical, environmental, lifestyle, and genetic information to understand human health and disease. Further details can be found at https://biobank.ndph.ox.ac.uk/showcase/.

To generate the case and control cohorts for the GWAS, we leveraged electronic health record (EHR) data from the UKB. The following data fields from the UKB were used to extract the patients: ICD-10 diagnoses (UKB Data Field 41270), ICD-9 diagnoses (UKB Data Field 41271), main ICD-10 diagnoses (UKB Data Field 41202), secondary ICD-10 diagnoses (UKB Data Field 41204), main ICD-9 diagnoses (UKB Data Field 41203), secondary ICD-9 diagnoses (UKB Data Field 41205), ICD-10 primary cause of death (UKB Data Field 40001), ICD-10 secondary cause of death (UKB Data Field 40002), Read-2, and Read-3 codes. The following codes are used for building the cohorts: ICD9: 332; ICD10: G20; Read2: 'F12..','F12z.';

Read3: 'F12..','.F22.','.F221','F120.','.F22Z'. To create the control cohorts, the following exclusion criteria were applied to the entire population: individuals with chromosome X aneuploidy (UKB Data Field 22019), genetic outliers (UKB Data Field 22027), self-reported parental history of Alzheimer's disease and PD (UKB Data Fields 20107 and 20110), self-reported medications (UKB Data Field 20003, Codes: 1141202024, 1141153490, 1141195974, 1140867078, 1140867494, 1141171566, 2038459704, 1140872064, 1140879658, 1140867342, 1140867420, 1140882320, 1140872216, 1140910358, 1141200458, 1141172838, 1140867306, 1140867180, 1140872200, 1140867210, 1140867398, 1140882098, 1140867184, 1140867168, 1140863416, 1140909802, 1140867498, 1140867490, 1140910976, 1140867118, 1140867456, 1140928916, 1140872268, 1140867134, 1140867208, 1140867218, 1140867572, 1140879674, 1140909804, 1140867504, 1140868170, 1140879746, 1141152848, 1141177762, 1140867444, 1140867092, 1141152860, 1140872198, 1140867244, 1140868172, 1140867304, 1140872072, 1140879750, 1140868120, 1140872214, 1141201792, 1140882100, 1141167976, 1140867820, 1140867948, 1140879616, 1140867938, 1140867690, 1141190158, 1141151946, 1140921600, 1140879620, 1141201834, 1140867152, 1140909806, 1140879628, 1140867640, 1141200564, 1141151982, 1140916288, 1141180212, 1140867860, 1140867952, 1140879540, 1140867150, 1140909800, 1140867940, 1140879544, 1140879630, 1140867856, 1140867726, 1140867884, 1140867922, 1140910820, 1140879556, 1141152732, 1140867920, 1140882244, 1140867852, 1140867818, 1141174756, 1140867916, 1140867888, 1140867850, 1140867624, 1140867876, 1141151978, 1140882236, 1140867878, 1201, 1140882312, 1140867758, 1140867712, 1140867914, 1140867944, 1140879634, 1140867756, 1140867934, 1140867960, 1140916282, 1141200570, 1141152736, 1141182732, 1141171578, 1141167690, 1141150834, 1141150840, 1141167700, 1140872320, 1140910516, 1141189132, 1141164060, 1140928274, 1140879648, 1140879668, 1141169700, 1140883514, 1140909816, 1140879644, 1140872420, 1140872434, 1141164872, 1141164068, 1140928338, 1141152896, 1141189134, 1140872348, 1141169666, 1140872440, 1140872378, 1140872460, 1140928014, 1140928010, 1141172616, 1141172620, 1141165546,

1141167618, 1141189254, 1141189256, 1140868364, 1141150596, 1141188640, 1141188642, 1141152904) for Alzheimer's treatment, PD, antipsychotic, or antidepressants, self-reported noncancer illness (UKB Data Field 20002, Codes: 1259,1260,1261,1262,1263,1264,1289, 1291,1433,1434,1531) in neuropsychiatric, neurodegenerative, and neuroimmunology disorders (covering motor neuron disease, myasthenia gravis, multiple sclerosis, PD, dementia/Alzheimer's/cognitive impairment, epilepsy, schizophrenia, mania/bipolar disorder/manic depression, cerebral palsy, other neurological problems, post-natal depression), any diagnosis of recurrent depression, chapter 6 of ICD-9 & -10 codes (Diseases of the Nervous System), and chapter 5 of ICD-9 & -10 codes (mental and behavioral disorders). All the individuals are of European ancestry.

## FinnGen data

The Release 9 of the FinnGen GWAS summary statistics was used in this study. The code used for the PD GWAS used is G6. Applications to access individual-level data in FinnGen can be made through the Finnish Biobanks' "FinnBB" portal (https://finbb.fi/) and summary GWA data can be accessed through the FinnGen website (https://www.finngen.fi/en/access_results).

## Genetic association analyzes

GWAS was performed using REGENIE v2.2.4. To account for population structure, a two-step procedure was used[30]. Briefly, in the first step, a whole genome regression model was fit on the genotype data using a leave-one-out chromosome (LOCO) scheme. The following covariates were used in the analysis: sex, year of birth, and the top 10 principal components of ancestry. In the second step, the LOCO predictions were used to perform variant association analysis using standard linear regression on the imputed array data. To account for the imbalance between the case and control numbers, the Firth approximation was used, and relatedness has been accounted for implicitly by REGENIE. The following criteria were used to include high-quality genotyped variants: minimum allele frequency (MAF) > 0.1%, genotyping rate > 0.95, Hardy-Weinberg equilibrium (HWE) test $P > 10^{-6}$, bin size in REGENIE = 400. Post-GWAS QC criteria include remove markers with MAF > 0.0001, imputation quality score (INFO) > 0.8, filter out variants with missingness > 5%. CMplot[88] was used to generate the Manhattan plots. Meta-analysis was performed using METAL[89] through a classical approach by computing weight effect size estimates using the inverse of the corresponding standard errors and a standard genome-wide multiple testing correction at $P < 5 \times 10^{-8}$ was applied. Pathway enrichment analysis was done using WebGestalt[90] using the Olink© protein panel used in the UKB-PPP as the reference. KEGG[38] biological pathways implemented in WebGestalt were used for enrichment analysis and pathways with FDR < 0.05 were called significant. ANCOVA was run on R 4.1.1 using age, sex, and age*sex as the covariates for comparing the LGALS3 abundance between different genotypes. Fine-mapping was performed using susieR v.0.12.35[91]. For fine mapping, a 500 kb window flanking independent SNPs were used and LD matrix was created using 1000 Genomes phase 3 reference panel[92].

## pQTL analyses from UKB-PPP

We leveraged pQTL summary statistics for 2923 proteins across 54,219 individuals from the UKB-PPP consortium data, described in ref.[14]. Briefly, REGENIE v2.2.1[30] was implemented via a two-step procedure to account for population stratification. In the association model several covariates in the discovery cohort have been used as follows: age, age², sex, age*sex, age²*sex, data batch, UKB center, UKB genetic array, time between blood sampling and measurement and the first 20 genetic principal components. A threshold of 5e-8 is used to define an association as statistically significant.

## Network analysis

To conduct network analysis, we used PPI data from PICKLE v3.3[29] where the cross-checked PPI network was selected. To conduct hypothesis-free network analysis, we used dmGWAS 3.0[46] with the order of neighbor genes d = 2 and cut-off increment rate r = 0.1 as recommended in ref.[29]. Networks were visualized using Cytoscape 3.9.1[93].

## Mendelian randomization and colocalization analysis

Two sample MR implemented in the R package TwoSampleMR v0.5.7[94,95] was utilized to identify causal plasma proteins in PD. Significant cis-pQTLs P < 5e-8 were used. Shared SNPs in both cis-pQTLs and PD GWAS from Nalls et al. paper were harmonized and clumped using linkage disequilibrium threshold of $r^2 < 0.001$. In cases where only one instrumental variable was available Wald ratio test was used. Testing for horizontal pleiotropy was performed using MRPRESSO v1.0 R package[96].

Colocalization analysis was performed using the R package coloc v5.2.3[97]. We used default priors of $p_1 = 10^{-4}$, $p_2 = 10^{-4}$, and $p_{12} = 10^{-5}$ where $p_1$ is the prior probability of a SNP being associated with PD, $p_2$ represents the prior probability of a SNP being associated with pQTLs, and $p_{12}$ is the prior probability of a SNP being associated to both PD and plasma pQTL. PP.H4 > 0.75 was considered as a strong colocalization, where PP.H4 represents the posterior probability of a shared SNP being associated with PD and pQTL-associated protein.

## Reporting summary

Further information on research design is available in the Nature Portfolio Reporting Summary linked to this article.

## Data availability

Summary statistics of the meta-analysis is deposited in GWAS Catalog under accession ID GCST90319903. This research has been conducted using the UK Biobank SNP Array Data and proteomic data under application numbers 52293 and 65851. The FinnGen R9 GWAS summary statistics on PD were used for meta-analysis and are available here [https://risteys.finregistry.fi/endpoints/G6_PARKINSON]. PLAC-seq data as obtained from Nott et al.[51] and scRNA-seq data was obtained from Kamath et al.[56]. Source data are provided with this paper.

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

## Acknowledgements

We want to acknowledge the participants and investigators of the UK Biobank and the FinnGen study.

## Author contributions

A.D.T. conceived the study design, developed the pipeline, conducted the computational analyses, and drafted the manuscript. D.T.T., C.D.W., L.H., and B.S. critically reviewed the manuscript and edited the manuscript. S.L. supervised the work and critically reviewed the manuscript.

## Competing interests

A.D.T., D.T.T., L.H., B.S., C.D.W., and S.L. are full-time employees of Janssen R&D and are stockholders of Johnson & Johnson.
