## [Peer Review File · Nature Communications]

REVIEWER COMMENTS

Reviewer #1 (Remarks to the Author):

The manuscript "Proteogenomic network analysis reveals dysregulated mechanisms and potential mediators in Parkinson's disease" is a well written article with relevance in the Parkinson's disease area.

The authors made a smart use of UK Biobank and FinnGen data to run a GWAS, determine pQTLs, run a pathway analysis based, a protein-protein interaction network analysis and a nice deep dive exploring single cell expression data and chromatin-loop interactions. The focus is finally set on Galectin 3, which is an interesting candidate for PD risk and progression as indicated by different lines of evidence. While yet more evidence would be required in order to take these findings to a target testing level, these are very interesting results in the field of Parkinson's disease, nicely backed up from several different research angles.

I have some comments on the approach and the methodology:

1. It seems to me that the authors are missing the opportunity of a formal causal analysis with Mendelian randomization, although they have all data required to do so.

2. The fact that protein levels are measured in plasma and not in CSF is a clear limitation in any study of a neurological disease. CSF proteins that do not cross the hematoencephalic barrier cannot be measured, and from those that cross the barrier, some will be diluted to noise levels and others will be confused with analogous processes taking place elsewhere in the body (e.g. neuroinflammation vs other types of inflammation). Arguably, pQTLs are not necessarily related to any particular disease status or matrix, but some pQTLs will be easier to detect given a certain disease status and a disease-relevant matrix. Analyses that use CSF instead of plasma as a matrix for pQTL analysis in Parkinson's disease have been published. A systematic comparison between those and the ones detected here, only for the proteins common to both approaches and only for the cis-pQTLs, could help putting context. CSF-based pQTLs will probably be mostly included in the ones detected here, and the differences will be mainly due to higher statistical power in the plasma analysis. But if the pQTLs present in the CSF analysis and absent in the plasma analysis are enriched for Parkinson's disease relevant pathways, that would point to a blind spot in the approach based on plasma.

3. It is not clear why there is a genetic analysis on UK Biobank only and then a meta-analysis adding FinnGen data. Also, how many SNPs analyzed? How many individuals in the FinnGen data set? Isn't the meta-analysis a better approach? And if not, why? Figure 1 gives a potentially false impression of coincidence between both analyses, which could simply reflect the fact that 1b includes 1a. For that matter it is also not clear to me why the authors would prefer to perform this meta-GWAS instead of using the one described in Nalls et al. 2019, which is clearly better powered.

4. It is not described how control for multiple testing is applied to this scenario with cis and trans pQTLs, where the number of hypotheses tested easily exceeds $1e10$. Is the significance threshold set at $5e-8$ for pQTLs as was for GWAS? The legend in figure 3a suggests \log_{10} p values {BTW, it should probably be $-\log_{10}$ p values, instead} in the order of magnitude of 3000, so $1e-3000$. Is that correct? If so, is that the lowest p-value for that position? Or is this the number of pQTLs for that position in the matrix rather than the $-\log_{10}$ p value? Also, the distinction between cis and trans used here doesn't sound very orthodox. Usually, we consider cis only when adjacent to the gene. A regulatory element can be trans even if in the same chromosome. A common choice is defining a cis-pQTL variant as a SNP located at $<1\text{Mb}$ upstream or downstream of the transcription start of the corresponding gene.

5. Out of the top 10 significantly enriched pathways, the first 6 do not look very meaningful in the context of Parkinson's. Don't the authors want to comment on that?

6. The workflow is clear, but it is not always clear what is the added benefit of applying each step in it. Lines 81 to 108 could benefit from highlighting the "why" alongside the "what".

7. Given that not the whole proteome, but only $\sim 3\text{K}$ proteins are directly measured, it could be interesting to know whether the number of pQTL-associated proteins in the network modeling is within or above the number expected by chance.

8. Are the differences in LGALS3 by rs11158026 genotype significant when pulling controls, incident cases and prevalent cases together? It could be shown as additional horizontal lines in Fig 7d.

Minor comments:

101: consider rewording "disrupted" proteins ("disrupted expression" or "dysregulated"?)

129: there is a difference between a "candidate" for PD association and a therapeutic "target". The latter sounds like an overstatement here.

166: "we narrowed down" -> how exactly?

175:177 Trans are necessarily more frequent than cis. The paragraph might be superfluous.

220:221 The color coding is better described in the figure's legend than in the main text.

233: consider rewording "disruption" ("expression disruption" or "dysregulation"?)

381: "marginally significant" for a $p=0.004$ seems a weird wording, is there a typo in the p-value?

429: avoid redundancy: "its potential as a" [-potential] "therapeutic target".

709: Figure 1, first box, third bullet, second line: revise the wording, it's not clear that the first number refers to nr of patients

709: Figure 1, first box, how many patients on FinnGen R9?

709: Figure 1, third box: rename to "Epigenomic Analysis", since "Omics Analysis" applies to all three boxes.

780: Figure 5, it would be nice to have the same color coding for GWAS in 5a and 5b.

798: Figure 6 The proteins listed in 6a do not correspond to the union or the intersection of Tables 1 and 2. Neither do the proteins listed in 6b,c,d correspond to those described in line 301 or in lines 204-205. Why these particular subsets?

818: Figure 7d, the color code is very confusing. The colors used in 7a,b,c mean something entirely different in 7d.

Reviewer #2 (Remarks to the Author):

The manuscript reports use of a three-stage system-level analysis framework to identify potential therapeutic targets for Parkinson's disease (PD). This is an area of considerable importance given the unmet need of the disease and the advances and scale of omics data that enable genuinely integrative analyses. The authors present a considerable body of work and data that highlight some links to PD. Below, I outline several comments that I think warrant consideration and may benefit the current work and the insights that are drawn from it.

-Reviewing the abstract, which outlines the aims and key findings raises some areas that may benefit from clarification.

o Overall, I think additional clarity could be offered on the aims of the study, the approaches used to address them, and what insights can be drawn. For example, the study claims to be the first to characterize “genetically-driven protein disturbances associated with PD”, but I am a little unclear exactly what is shown in this regard and how it should be interpreted – see below in the review for some more specific questions.

o The study reports “577 pQTL-associated proteins, which enrich for PD-related pathways such as cytokine receptor interactions and lysosomal function”. Could the authors clarify what defines a “pQTL-associated protein”(this is used throughout the paper) and what it means that they enrich for PD-related pathways? Also, in the abstract, clarify the significance of the 9 proteins mentioned – it simply states that they were identified.

- In stage 1, genetic and proteomic data were used with GWAS and pQTL methods to identify PD-associated genes and proteins. GWASes of PD were conducted using UK Biobank and FinnGen data, followed by a meta-analysis of UKB and FinnGen GWAS results. The results of the meta-analysis were then overlapped with pQTL data from the UK Biobank Pharma Proteomics Project to identify proteins that associate with genetic variation that also associates with PD risk. This simple intersection of PD-associated SNPs with plasma pQTLs is likely to be confounded by LD. It is unclear if any colocalization of disease and proteomic signals was required – could the authors clarify? I'd suggest that it would be valuable to do so, given the reported extent of confounding by LD.

o In the results, the authors note that 577 proteins (~20% of the total panel) showed some link. Thus, I don't think these are necessarily "PD-associated proteins".

o This feels fundamental, as the additional analyses such as network modelling are built on these results.

- Further, the authors report that they restricted the GWAS sample size to "to establish the associations between genome using orthogonal data sets generated from the same samples and proteome", but it is unclear the extent to which samples were orthogonal (I think there will be overlap?) and what benefit this brings? For example, it drastically reduces the extent of genetic signal for PD. And, the use of FinnGen data seem inconsistent with this statement. I don't understand the logic of the decision, especially because the analytical approaches used are two-sampled in nature. Perhaps the authors could qualify the additional insights gained through this approach?

-The authors variably refer to a wish to identify targets, which suggests that they wish to identify proteins with a causal link to disease – is this correct, and is this what is being suggested for the implicated proteins? Did the authors attempt to perform any form of causal analyses for these proteins? For example, for LGALS3 have other pQTLs and are they associated with PD? The authors cite previous MR analyses with what appear to be nominally significant findings – could the authors address the question using these expanded data?

-The term "mediator" is also used throughout (alongside "indirect mediator"), which isn't particularly clear to this reader on what relationship they are suggested to mediate. Perhaps the authors could consider the use of directed graphs throughout to outline their hypotheses, as per ST-2.

-Again, as I read the conclusions, some of the statements were unclear on what exactly they referred to. E.g. "Our study is the first effort to leverage proteogenomic UKB data in a network framework to characterize the genome-proteome relations and model their interactions."

In addition, there are a number of other more general or minor comments

-Generally, the text would benefit from being more concise – for example, the results section starts with over a page of additional exposition. Could this be shortened or interspersed with the relevant section of actual results?

-Results, paragraph 2 – confirm which analyses were performed in step 1 – PD, I assume?

-Please provide citation for the public PPI network in the results when first mentioned.

-“For the remaining nominated proteins, we did not observe published studies to demonstrate their links with neurodegenerative disorders.” – the next sentence then reports a number of links for LGALS3.

-In the hypothesis-driven network analysis were the significant loci from the meta-analysis mapped to specific genes/proteins through the MAGMA gene-based scores? Same for the differential expression analyses of PD-associated loci with single-cell RNAseq data. It's unclear from the current text.

Reviewer #3 (Remarks to the Author):

Reviewer #4 (Remarks to the Author):

This study performs an integrative analysis across complementary ‘omics datasets, that include recent proteomics results (UKB-PPP) that are anchored to genetic data from UKB to derive disease mechanisms and potential disease mediators of Parkinson’s disease (PD). Through these analyses the authors highlight PD-related pathways and proteins including LGALS3 for further study. I felt the paper was well written and showcases how population proteomics data can be used with

orthogonal datasets to triangulate on genes with therapeutic potential in Parkinson's disease that might not be obvious from GWAS alone.

Major Comments.

It wasn't clear to me what the benefit of conducting a separate PD GWAS compared to Nalls et al. was given the drop in case size and thus presumably power. The authors make a justification that their objective was to establish associations between the genome using orthogonal datasets generated from the same samples and proteome (page 5 line 126), however it was unclear what this overlap was, for example how many of the samples in the GWAS also had proteomic samples available in UKB?

When integrating proteo-genomics results with PD it appears that a simple intersect was used (page 5 line 139), was there a reason that a more sophisticated approach such as colocalization (e.g. coloc) was not attempted? My concern is that with a simple intersect approach whilst a variant may be associated with both a protein's abundance and PD the underlying causal variant could be different which may generate unreliable results for downstream analyses. This information would be especially useful for the 'deep dive' on rs11158026-T and LGALS3.

There are very sparse details on pathway enrichment method (page 6 line 151), for example was the fact that the selection of genes/proteins was limited to those assayed in UKB PPP considered? How were the dependencies between genes due to LD accounted for? Without these details hard to assess the relevance of this analysis.

I wasn't entirely clear on the SNP-to-protein specificity section (page 6 line 163); how was LD considered etc. (especially given the Table 2 contains HLA region). Is the observation that there are more trans associations than on the same chromosome surprising (page 6 line 175)? Given that the number of potential cis pQTL variants for a protein is an order of magnitude less than for trans/different chromosome pQTLs?

I thought hypothesis free PPI network analysis was fine as MAGMA was used to score genes which I believe makes some attempt to consider collinearity between GWAS stats due to LD. I found the hypothesis driven network (page 7 line 219) analysis harder to assess, given the strong assumptions (previously mentioned) that the closest gene is causal and therefore used to construct the network. For permutation analysis I wasn't convinced on the empirical distribution as the ~3K proteins assessed via UKB PPP are likely to be non-random in their selection, perhaps enriched for those with clinical application/association with disease, could that be an explanation for the enrichment?

For the chromatin loop analysis, which was interesting, I think fine mapping would have greatly increased the value of this analysis as this would provide some posterior estimate of a variant being causal. Whilst I understand the focus on brain tissue types, for context I think it worth looking across loop data for blood (given that proteomics was collected in plasma) to frame how brain specific the proposed regulatory variants are. This could bolster the point in the discussion on the relevance of plasma proteomics to neurodegenerative disease (somewhat undermined by focus on brain specific transcriptomic and epigenomic datasets – in my opinion).

Page 11 line 345 does the association with rs11158026-T in PD and with LGALS3 colocalise such they are likely due to the same causal variant? I suspect that they do given the MR analysis referred to later on in this section but be good to have clarity.

I was a bit baffled by the LGALS3 incident / prevalence association analyses with genotype (page 11 line 363) would not a Cox regression type approach be better suited to analysing incident cases to better understand the protective effect of rs11158026-T?

The methods seem quite sparse as mentioned enrichment analysis is hard to interpret without a better understanding as to what was done. There are no methods for for single cell analysis (page 8 line 262) e.g. it says that significantly upregulated genes for each cluster were extracted but doesn't say how.

In terms of data availability – the GWAS summary statistics should be submitted to a public repository.

Minor comments

Could the authors provide a qq plots/ an estimate for genomic control for their GWAS to provide reassurance that there is no confounding.

For Table1 and Table2

-can you add to legend what the listed allele is (I assume the effect allele) but good to make explicit.

-could the gene column be renamed 'Closest Gene', also I think helpful to report MAF (for Finngen also for table2, so reader can assess founder effects etc.).

Perhaps worth mentioning that 12-40340400-G-A(rs34637584) given its rarity is also replicated using WES data (from Gene Bass).

Outside of LRRK2/APOE the associations appear to be non-coding common variants? Given there was no attempt to fine map and the fact that an index variant is not necessarily causal how relevant are V2G, CADD etc. columns?

Page 7 line 201 – what was the justification for top 1% was there an elbow in the scores or was this purely a pragmatic heuristic?

Page7 line 204 what is the expectation of the number of shared proteins between the two groups i.e. does the observation of 14 proteins exceed that which we would expect by chance? I believe it does but good to have clarity

In suppl table 4 it would be helpful to add a column that details the genes within GWAS locus (red fig 5a) or with an intersecting pQTL.

Page7 line 210 abbreviation for AD is not previously defined suggest replacing with Alzheimer Disease.

Page 7 line 211-215 what is the genetic correlation between AD, PD and LBD? This might help a reader understand whether this degree of overlap is expected in terms of proteins/genes with putative shared aetiology.

Figure 1b – I see that chr23 is missing from the meta-analysis and there may be technical reasons for this. Given that there aren't any associations for chr23 it might be worth omitting from panel 'a' as well – in this way easier for reader to line up EHR GWAS with meta-analysis with Finngen.

Figure 1c – I am not sure this adds much I would suggest removing – as there is no explanation of how p-values were calculated (tricky to do in the presence of LD).

Fig 3a What are the units of scale on the x and y axis, seems to be chromosome? For example, is the line of trans associations near 5 on the x axis due to HLA association?

Fig 3b As previously mentioned be great to see which of these actually colocalise rather than just intersect.

Figure 5a – consider using a different colors, red green not color-blind friendly also blue for pQTL associated proteins which are PD associated GWAS loci, is a bit confusing as default/background appears to be light blue.

Responses to Reviewers' Comments

Summary of changes

We would like to thank the editor and the reviewers for your evaluation of our manuscript (NCOMMS-23-34002) titled 'Proteogenomic network analysis reveals dysregulated mechanisms and potential mediators in Parkinson's disease'. We have carefully addressed all of the comments raised by the editor and the reviewers and made modifications to the text. Please find below our point-by-point responses to the reviewers' specific comments. All page numbers refer to the revised manuscript file, and relevant sections have been yellow highlighted.

Reviewer 1

The manuscript "Proteogenomic network analysis reveals dysregulated mechanisms and potential mediators in Parkinson's disease" is a well written article with relevance in the Parkinson's disease area.

The authors made a smart use of UK Biobank and FinnGen data to run a GWAS, determine pQTLs, run a pathway analysis based, a protein-protein interaction network analysis and a nice deep dive exploring single cell expression data and chromatin-loop interactions. The focus is finally set on Galectin 3, which is an interesting candidate for PD risk and progression as indicated by different lines of evidence. While yet more evidence would be required in order to take these findings to a target testing level, these are very interesting results in the field of Parkinson's disease, nicely backed up from several different research angles.

I have some comments on the approach and the methodology:

Author's Response: We appreciate the reviewer's positive comments on the general interest to the field. We have now addressed the reviewer's additional comments below.

1. Reviewer's comment: *It seems to me that the authors are missing the opportunity of a formal causal analysis with Mendelian randomization, although they have all data required to do so.*

Author's Response: Thank you for raising this important point. We have now conducted a Mendelian Randomization (MR) analysis. In addition, we have performed a colocalization analysis on the identified potential causal proteins from MR to generate further evidence about putative causal association of the proteins derived from MR. Moreover, we performed colocalization analysis on the loci derived from the UKB-FinnGen meta-analysis. We have added a new section to page 8 and created Table 3 for the MR and colocalization results. Moreover, the details of the implemented methods are provided in Methods and Materials section. Finally, we have further discussed the

findings in the Discussion section. The added paragraph on page 7 is as follows: “We conducted a Mendelian Randomization (MR) analysis to identify potential PD causally-linked proteins. We used 6,381 identified cis-pQTLs from the UKB-PPP as the instrumental variables and protein abundance as the Exposure in this analysis (Supplementary Table 1). We took advantage of summary statistics from Nalls et al. [37] to use in the MR analysis as the outcome (Methods and Materials). We identified a total of 122 proteins showing a significant association at $FDR < 0.05$. Following MR, we conducted a colocalization analysis between the UKB-FinnGen meta-GWAS and pQTLs from the UKB-PPP to characterize if meta-analysis signals colocalize with pQTL hits in the MR-derived proteins aimed at assessing whether the genetic factors from meta-GWAS also colocalize with the genetic predictor of the protein abundances for supporting the MR findings which are based on the findings of Nalls et al. We found significant colocalization signals for 18 proteins identified in MR with a colocalization posterior probability > 0.9 . These 17 proteins are shown in Table 3 and the full list of all MR results is provided in Supplementary Table 1. In addition, we ran a colocalization analysis on the UKB-FinnGen meta-analysis genes (Table 2). Among which HIP1R (H4 posterior probability, $H4.PP=1$), GCH1 ($H4.PP=0.99$), SNCA ($H4.PP=1$), TMEM175 ($H4.PP=0.95$) show a significant colocalization signal”.

2. Reviewer’s comment: *The fact that protein levels are measured in plasma and not in CSF is a clear limitation in any study of a neurological disease. CSF proteins that do not cross the hematoencephalic barrier cannot be measured, and from those that cross the barrier, some will be diluted to noise levels and others will be confused with analogous processes taking place elsewhere in the body (e.g. neuroinflammation vs other types of inflammation). Arguably, pQTLs are not necessarily related to any particular disease status or matrix, but some pQTLs will be easier to detect given a certain disease status and a disease-relevant matrix. Analyses that use CSF instead of plasma as a matrix for pQTL analysis in Parkinson’s disease have been published. A systematic comparison between those and the ones detected here, only for the proteins common to both approaches and only for the cis-pQTLs, could help putting context. CSF-based pQTLs will probably be mostly included in the ones detected here, and the differences will be mainly due to higher statistical power in the plasma analysis. But if the pQTLs present in the CSF analysis and absent in the plasma analysis are enriched for Parkinson’s disease relevant pathways, that would point to a blind spot in the approach based on plasma.*

Author’s Response: Thank you for raising this important comment. Based on your recommendation, we made a systematic comparison between cis-pQTLs in plasma and CSF followed by adding a new section to pages 8-9 reading: “Circulating plasma proteins to study neurological diseases are constrained by the absence of proteins that do not cross the blood-brain barrier (BBB). One alternative tissue for neurological studies is Cerebrospinal Fluid (CSF). Therefore, a systematic comparison between plasma-derived protein measurements versus CSF can potentially shed light on proteins that cannot be captured in plasma either because they cannot make it through

the BBB or can make it through at lower concentrations leading to their dilution to noise when preprocessing the data. Leveraging pQTL associations derived from CSF by Kaiser et al. [21], we made a systematic evaluation using our identified cis-pQTLs and their associated proteins. Please note that there are systematic differences between the proteomic data used in our study as opposed to the data used by Kaiser et al. [21]. UKB-PPP proteomics data is generated by the Olink Explore 3072 antibody-based proximity extension assay (PEA), measuring 2,941 protein analytes capturing 2,923 unique proteins. On the other hand, the data used by Kaiser et al. [21] is generated by SomaScan aptamer-based proteomics platform [28] which captures 4,135 proteins. The two platforms share a total of 1,583 unique proteins. Due to technical differences of data processing on the two platforms as well as the protocols used for characterizing pQTLs, not all the pQTLs might have been shared between the two studies.

The CSF-based study reports significant cis pQTLs for 856 Off-rate Modified Aptamers (SOMAmers) corresponding to 744 unique proteins [21]. Among these proteins, 474 proteins are also profiled in the UKB-PPP data. Comparing the CSF-derived pQTL-associated proteins with our cis-pQTL associated proteins, 151 proteins are shared between the two (Supplementary Table 1) leading to a total of 323 proteins that are found to show significant cis-pQTL associations only in CSF. We conducted a pathway enrichment analysis on significant proteins unique to CSF using KEGG gene sets [29]. Only one pathway 'Complement and coagulation cascades' (FDR=0.002) passed the significance threshold of FDR<0.05. We had not identified this pathway to be significant in our analysis based on the pQTL-associated proteins in plasma. The complement system is a tightly regulated innate immune system playing a key role in regulating the normal function and development of the central nervous system (CNS) [30]. According to Fatoba et al. [30], accumulating evidence from human postmortem studies demonstrate that neuronal and glia cells are capable of expressing a majority of the complement molecules in CNS. This pathway has been reported in several CNS-related studies such as multiple system atrophy [31], brain proteome profiling of PD in humans and mice [32-34], and psychosis [35].

In addition to evaluating the pQTL-associated proteins in both datasets, we also compared the cis-pQTLs that were identified in both studies. Two pQTLs were shared on both studies including rs429358 and rs557011. In CSF, the missense variant rs429358 was significantly associated with three proteins APOC2 (Beta=0.89, P=3.56E-39), APOE (Beta=0.82, P=2.49E-45) and CEACAM19 (Beta=0.46, P=4.05E-17). However, this variant was only significantly associated with APOE in plasma with a negative direction of effect (Beta=-1.01, -log₁₀(P)=2069.74). rs557011 was associated with HLA-DQA2 in CSF (Beta=0.53, P=1.72E-42) while being significantly associated with HLA-DRA in plasma (Beta=0.37, -log₁₀(P)=455.6). The main difference observed is the opposite association of rs429358 with APOE in plasma versus CSF.

Our observation indicates the intrinsic limitations of measuring circulating plasma proteins to study brain diseases and need for leveraging CNS-specific tissues to shed light on genome-proteome aberrations in PD that may not be captured in plasma.”

3. Reviewer's comment: *It is not clear why there is a genetic analysis on UK Biobank only and then a meta-analysis adding FinnGen data. Also, how many SNPs analyzed? How many individuals in the FinnGen data set? Isn't the meta-analysis a better approach? And if not, why? Figure 1 gives a potentially false impression of coincidence between both analyses, which could simply reflect the fact that 1b includes 1a. For that matter it is also not clear to me why the authors would prefer to perform this meta-GWAS instead of using the one described in Nalls et al. 2019, which is clearly better powered.*

Author's Response: Thank you for raising this point. Given that the total number of PD samples in the UKB GWAS is limited, we further moved forward and meta-analyzed our results with FinnGen R9 to increase our statistical power. In addition, given the concordance between the genetic data and the proteomics data which are orthogonal datasets on the same set of individuals, our intention was to leverage this harmony between the two data for downstream analyses. In addition, conducting hypothesis-free network analysis requires having GWAS results for all the SNPs to calculate gene-based scores across the entire genome. This is not feasible by using the summary statistics from Nalls paper as it only provides the top significant hits not the entire list of evaluated SNPs across the genome. We totally agree with your comment that the meta-analysis by Nalls et al. is significantly well powered given the total number of samples that it covers. Therefore, to address your point, we have now used the significant SNPs reported by Nalls et al. and analyzed it in our set of pQTLs as well as their associated proteins in the UKB-PPP. Based on this, the following changes have been made to the text as follows:

The total number of SNPs as well as the FinnGen sample size and the overall sample size of the UKB-FinnGen are incorporated to page 5 paragraph 3 reading: "FinnGen R9 contains 4,235 PD cases and 373,042 healthy controls leading to a total 7,099 PD cases and 531,918 healthy controls in this study. Overall, 11,114,080 SNPs shared between the two datasets were evaluated for meta-analysis".

We also analyzed the significant SNPs provided by Nalls et al. and examined them if any of them are pQTLs in the UKB-PPP. Later, we conducted a pathway enrichment analysis and discussed the similarities between the enriched pathways with the pathways that were identified through analyzing the UKB-FinnGen results. The following is added to the text in page 7 paragraph 1: "The UKB-FinnGen meta-analysis is restricted in terms of the sample size. Therefore, we sought to leverage the summary statistics provided by Nalls et al. [1] and evaluate their potential associations with plasma protein concentrations in the UKB-PPP. 107 SNPs provided by Nalls et al. were evaluated. 51 SNPs were pQTLs in the UKB-PPP associated with a total of 204 proteins leading to a total number of 287 pQTL-protein associations (Supplementary Table 1). Out of 204 identified proteins, 149 proteins were shared with the 577 pQTL-associated proteins from UKB-FinnGen while 55 proteins were not identified before. pQTL-associated proteins are the proteins whose abundance is significantly associated with the SNPs identified in the UKB-FinnGen meta-analysis. We conducted a pathway

enrichment analysis on the 204 proteins leading to a total of 6 significant pathways (Supplementary Table 3). Among which, three pathways were shared with the enriched pathways from the PD associated proteins from UKB-FinnGen including: lysosome (FDR=4.62E-4), Graft-versus-host disease (FDR=0.003), and Cell adhesion molecules (CAMs) (FDR=0.03). the other three pathways that are unique to this analysis include: Antigen processing and presentation (FDR=2.88E-5), Staphylococcus aureus infection (FDR=3.94E-4), and Natural killer cell mediated cytotoxicity (FDR=0.03).”

With respect to your comment about the figure, we have now modified the middle panel in Fig. 1 to create a clear separation between each panel in the figure to avoid any potential confusions.

4. Reviewer’s comment: *It is not described how control for multiple testing is applied to this scenario with cis and trans pQTLs, where the number of hypotheses tested easily exceeds $1e10$. Is the significance threshold set at $5e-8$ for pQTLs as was for GWAS? The legend in figure 3a suggests $\log_{10} p$ values (BTW, it should probably be $-\log_{10} p$ values, instead) in the order of magnitude of 3000, so $1e-3000$. Is that correct? If so, is that the lowest p-value for that position? Or is this the number of pQTLs for that position in the matrix rather than the $-\log_{10} p$ value? Also, the distinction between cis and trans used here doesn't sound very orthodox. Usually, we consider cis only when adjacent to the gene. A regulatory element can be trans even if in the same chromosome. A common choice is defining a cis-pQTL variant as a SNP located at <1Mb upstream or downstream of the transcription start of the corresponding gene.*

Author’s Response: Thank you for raising this important comment. We have used a significance threshold of $5e-8$ for pQTL associations as was for GWAS. We have now indicated this in the text on page 18 paragraph 2 reading “A threshold of $5e-8$ is used to define an association as statistically significant”. As you rightly put, we have now modified the legend of Fig. 3a to ‘ $-\log_{10}(P)$ ’. This legend refers to the smallest p-value in that position. We have also modified our definition of cis associations to align with the common definitions of such associations. Based on this, we clarified the definition of cis associations in page 7 paragraph 3 reading “The frequencies of cis associations (defined as ± 1 kb flanking regions of gene boundaries) ...”. In addition, we added one more column to the list of pQTL associations provided in Supplementary Table 1 where we indicate if each association is cis or trans based on our modified definition.

5. Reviewer’s comment: *Out of the top 10 significantly enriched pathways, the first 6 do not look very meaningful in the context of Parkinson's. Don't the authors want to comment on that?*

Author’s Response: Thank you for raising this important point. The data generated by the UKB-PPP is not dedicated to PD and the list of proteins in the Olink panel include low-abundant inflammatory proteins, proteins actively secreted into blood circulation,

approved and ongoing drug targets, organ-specific proteins leaked into blood circulation, and the proteins representing exploratory biomarkers. Therefore, not all the proteins implicated in PD have been profiled in this consortium. On the other hand, neuroinflammation plays a critical role in the pathogenesis of PD. In fact, central and peripheral inflammation play a vital role in pathological features of PD [1, 2]. Among inflammatory markers, chemokines such as chemokine ligand (CCL) and CXCL family as well as cytokines such as tumour necrosis factor (TNF) and interleukin (IL) are known as critical signaling molecules of immune activation in the central nervous system [3]. Looking into the enriched proteins provided in Supplementary Table 3, we can see that the top pathways are predominantly enriched by immune-related markers such as CCLs, ILs, and CXCLs. This is in line with our existing knowledge about the role of neuroinflammation in the PD pathogenesis. Moreover, PD is known to share several GWAS loci with multiple autoimmune disorders such as rheumatoid arthritis [4, 5]. This explains observing immune related pathways such as ‘Cytokine-cytokine receptor interaction’, ‘Rheumatoid arthritis’, ‘Chemokine signaling pathway’, and ‘Intestinal immune network for IgA production’. Taken together with the fact the UKB-PPP is not a PD-dedicated study which is designed to reflect the entire population in the UK Biobank, not all the PD-related genes and their encoded proteins have been profiled by the UKB-PPP. Therefore, we totally agree with your delicate point and to make it more clear to the readers, we have added additional explanation to page 6 paragraph 3 reading “In addition to known PD-related pathways, majority of the other significant pathways were enriched for inflammatory markers including chemokines such as chemokine ligand (CCL) and CXCL family as well as cytokines such as tumor necrosis factor (TNF) and interleukin (IL). ‘Rheumatoid arthritis’-related pathways were also predominantly enriched for inflammatory markers. This is in line with the existing knowledge of multiple shared genetic variants between PD and rheumatoid arthritis and several other autoimmune diseases [35, 36]. We point out that the UKB-PPP is a population-based study with proteomic evaluation based on a commercial panel of 2,923 proteins, therefore many known PD-related proteins were not profiled in the consortium. The proteins in the Olink panel include low-abundant inflammatory proteins, proteins actively secreted into blood circulation, approved and ongoing drug targets, organ-specific proteins leaked into blood circulation, and the proteins representing exploratory biomarkers. As a result, some known PD-related pathways may not show significant enrichment as the relevant proteins were not assayed”.

6. Reviewer’s comment: *The workflow is clear, but it is not always clear what is the added benefit of applying each step in it. Lines 81 to 108 could benefit from highlighting the "why" alongside the "what".*

Author’s Response: To address your comment regarding ‘why’ doing this type of analysis, we have added an opening part to page 4 paragraph 1 to further clarify the necessity of such an analytical approach as well as a few successful examples that leverage integrated methods. This newly added part reads: “Identifying causal genes that can be further considered as potential therapeutic targets remains a complex undertaking [24]. Characterizing causal variants that underpin genetic associations with a disease as well as elucidating the mechanistic impacts of genetic aberrations on the

human proteome and normal protein interaction necessitates integrative approaches to combine genetics with other omics data types such as proteomics and epigenomics [25, 26]. Moreover, since majority of drug targets are proteins, then proteome-based MR as well as modeling alterations in the human interactome caused by genetic abnormalities is of great importance in terms of identifying novel disease causal genes/proteins [27, 28]”.

7. Reviewer’s comment: *Given that not the whole proteome, but only ~3K proteins are directly measured, it could be interesting to know whether the number of pQTL-associated proteins in the network modeling is within or above the number expected by chance.*

Author’s Response: Thank you for raising this critical observation. We had done an empirical test to make sure that the overlap between the pQTL-associated proteins and the direct neighbors of PD-associated risk loci was not by chance. This evaluation is indicated in paragraph 2 of page 10 as follows: “We ran a set of permutation tests to check if the overlap between the pQTL-associated proteins and the direct neighbors of PD-associated risk loci was not by chance. We randomly selected proteins from the entire PPI network with a size equal to the network presented in Figure 5b and tested their overlap with the pQTL-associated proteins for 10,000 iterations. In each iteration, we tested the overlap using Fisher’s exact test and created an empirical distribution of the p-values. An empirical p-value of 0.014 was observed, denoting significant enrichment of pQTL-associated proteins within the direct neighbors of PD GWAS loci in human PPI network”.

8. Reviewer’s comment: *Are the differences in LGALS3 by rs11158026 genotype significant when pulling controls, incident cases and prevalent cases together? It could be shown as additional horizontal lines in Fig 7d.*

Author’s Response: we have now added those p-values to Fig. 7d as well as additional description to the caption of this figure.

Minor comments:

9. Reviewer’s comment: *101: consider rewording "disrupted" proteins ("disrupted expression" or "dysregulated"?)*

Author’s Response: We replaced “disrupted” with “dysregulated”.

10. Reviewer's comment: 129: *there is a difference between a "candidate" for PD association and a therapeutic "target". The latter sounds like an overstatement here*

Author's Response: We changed the wording to "candidate target identification" in page 5 paragraph 5.

11. Reviewer's comment: 166: *"we narrowed down" -> how exactly?*

Author's Response: We added "(n=10)" to the text to indicate the number of SNPs used here (page 7 paragraph 3).

12. Reviewer's comment: 175:177 *Trans are necessarily more frequent than cis. The paragraph might be superfluous.*

Author's Response: We removed this paragraph.

13. Reviewer's comment: 220:221 *The color coding is better described in the figure's legend than in the main text.*

Author's Response: We have removed the color description from the text. They have been described in the figure legend.

14. Reviewer's comment: 233: *consider rewording "disruption" ("expression disruption" or "dysregulation"?)*

Author's Response: We replaced "disruption" with "dysregulation" (page 10 paragraph 2).

15. Reviewer's comment: 381: *"marginally significant" for a $p=0.004$ seems a weird wording, is there a typo in the p-value?*

Author's Response: Thank you for noting this. There was a typo here. We have corrected the p-value to 0.04 (page 14 paragraph 2).

16. Reviewer's comment: 429: *avoid redundancy: "its potential as a" [-potential] "therapeutic target".*

Author's Response: This sentence is now corrected.

17. Reviewer's comment: 709: *Figure 1, first box, third bullet, second line: revise the wording, it's not clear that the first number refers to nr of patients*

Author's Response: We revised this figure and addressed this comment.

18. Reviewer's comment: 709: *Figure 1, first box, how many patients on FinnGen R9?*

Author's Response: We have included the cohort size on page 5 paragraph 3.

19. Reviewer's comment: 709: *Figure 1, third box: rename to "Epigenomic Analysis", since "Omics Analysis" applies to all three boxes.*

Author's Response: This figure is revised.

20. Reviewer's comment: 780: *Figure 5, it would be nice to have the same color coding for GWAS in 5a and 5b.*

Author's Response: We have now harmonized both panels.

21. Reviewer's comment: 798: *Figure 6 The proteins listed in 6a do not correspond to the union or the intersection of Tables 1 and 2. Neither do the proteins listed in 6b,c,d correspond to those described in line 301 or in lines 204-205. Why these particular subsets?*

Author's Response: Fig. 6a represents the list of differentially expressed genes, that were presented in tables 1-2, in the analyzed single-cell data in dopamine neurons. Fig. 6b-c is not related to the tables 1-2. These are the genes whose related proteins were shared between the hypothesis-free and hypothesis-driven network analyses. In fact, the panels b-d include the proteins indicated in line 301, whereas these panels are not related to the proteins indicated in lines 204-205.

22. Reviewer's comment: 818: *Figure 7d, the color code is very confusing. The colors used in 7a,b,c mean something entirely different in 7d.*

Author's Response: To avoid any confusion to the readers, we have overhauled this figure.

Reviewer 2

The manuscript reports use of a three-stage system-level analysis framework to identify potential therapeutic targets for Parkinson's disease (PD). This is an area of considerable importance given the unmet need of the disease and the advances and scale of omics data that enable genuinely integrative analyses. The authors present a considerable body of work and data that highlight some links to PD. Below, I outline several comments that I think warrant consideration and may benefit the current work and the insights that are drawn from it.

Reviewing the abstract, which outlines the aims and key findings raises some areas that may benefit from clarification.

Overall, I think additional clarity could be offered on the aims of the study, the approaches used to address them, and what insights can be drawn. For example, the study claims to be the first to characterize “genetically-driven protein disturbances associated with PD”, but I am a little unclear exactly what is shown in this regard and how it should be interpreted – see below in the review for some more specific questions.

1. Reviewer's comment: *The study reports “577 pQTL-associated proteins, which enrich for PD-related pathways such as cytokine receptor interactions and lysosomal function”. Could the authors clarify what defines a “pQTL-associated protein”(this is used throughout the paper) and what it means that they enrich for PD-related pathways? Also, in the abstract, clarify the significance of the 9 proteins mentioned – it simply states that they were identified.*

Author's Response: Thank you for raising this important point. ‘pQTL-associated proteins’ are the proteins whose abundance is significantly associated with the genetic variation SNPs identified through UKB-FinnGen meta-analysis. To clarify this definition, we added the following sentence to page 7 paragraph 1: “pQTL-associated proteins are the proteins whose abundance is significantly associated with the SNPs identified in the UKB-FinnGen meta-analysis”. Upon conducting a pathway enrichment analysis using these 577 proteins, several pathways were found to be enriched for these proteins such as some of the known PD-related pathways. By enrichment, we mean that some of the pQTL-associated proteins are a member of some known biological pathways (each pathway consists of a set of proteins) and these proteins are significantly overrepresented in these pathways more than would be expected by chance from the reference set of proteins. For further clarification about the pathway enrichment analysis, we have made additional explanations in page 6 paragraphs 3 and page 7 paragraph 2.

Moreover, we removed ‘pQTL-associated’ from the abstract to avoid any confusion for the readers. Also, we added the following sentence to the abstract to clarify the significance of these 9 proteins as follows: “... signifying their potential roles in the disease pathogenesis”.

2. Reviewer's comment: *In stage 1, genetic and proteomic data were used with GWAS and pQTL methods to identify PD-associated genes and proteins. GWASes of PD were conducted using UK Biobank and FinnGen data, followed by a meta-analysis of UKB and FinnGen GWAS results. The results of the meta-analysis were then overlapped with pQTL data from the UK Biobank Pharma Proteomics Project to identify proteins that associate with genetic variation that also associates with PD risk. This simple intersection of PD-associated SNPs with plasma pQTLs is likely to be confounded by LD. It is unclear if any colocalization of disease and proteomic signals was required – could the authors clarify? I'd suggest that it would be valuable to do so, given the reported extent of confounding by LD.*

Author's Response: Thank you so much for raising this important point. To address your comment, in this revision of the manuscript, we have conducted a thorough causal inference analyses including Mendelian Randomization and Colocalization analysis. This led us to identify putatively causal proteins in PD as well as additional evidence if GWAS-derived SNPs colocalize with the identified pQTLs. We provided the results in page 7 reading “We conducted a Mendelian Randomization (MR) analysis to identify potential PD causally-linked proteins. We used 6,381 identified cis-pQTLs from the UKB-PPP as the instrumental variables and protein abundance as the Exposure in this analysis (Supplementary Table 1). We took advantage of summary statistics from Nalls et al. [37] to use in the MR analysis as the outcome (Methods and Materials). We identified a total of 122 proteins showing a significant association at $FDR < 0.05$. Following MR, we conducted a colocalization analysis between the UKB-FinnGen meta-GWAS and pQTLs from the UKB-PPP to characterize if meta-analysis signals colocalize with pQTL hits in the MR-derived proteins aimed at assessing whether the genetic factors from meta-GWAS also colocalize with the genetic predictor of the protein abundances for supporting the MR findings which are based on the findings of Nalls et al. We found significant colocalization signals for 18 proteins identified in MR with a colocalization posterior probability > 0.9 . These 17 proteins are shown in Table 3 and the full list of all MR results is provided in Supplementary Table 1. In addition, we ran a colocalization analysis on the UKB-FinnGen meta-analysis genes (Table 2). Among which HIP1R (H4 posterior probability, $H4.PP=1$), GCH1 ($H4.PP=0.99$), SNCA ($H4.PP=1$), TMEM175 ($H4.PP=0.95$) show a significant colocalization signal”.

3. Reviewer's comment: *In the results, the authors note that 577 proteins (~20% of the total panel) showed some link. Thus, I don't think these are necessarily “PD-associated proteins”. This feels fundamental, as the additional analyses such as network modelling are built on these results.*

Author's Response: Thank you for this important observation. We agree that not all of these 577 proteins may be associated with PD. Therefore, we removed “PD-associated proteins” from the abstract and made a clarification in page 6 paragraph 1 reading “These proteins may not necessarily be disease associated, thus we further conducted causal inference analyses to characterize putative causal proteins”. The network

analysis, either hypothesis-free or hypothesis-driven, aims to take into account all the possible proteomic alterations associated with PD genetic variants. This is necessary in that if we use only proteins that are being associated with PD genetic variants which are also known to be disease associated, then the network modeling will be extremely biased and we will not be able to pinpoint new proteins that indirectly impact the disease risk or disease-related mechanisms such as LGALS3.

With respect to the network analysis, two approaches are taken, hypothesis-free and -driven. In the hypothesis-free approach, we do not use these 577 proteins, but we use a PPI network and gene-based scores generated from the UKB-FinnGen meta-analysis. The final product is a list of proteins that shape the top 1% of the most significant PD-associated network modules. These top modules, which include 175 proteins, are the outcome of integration of the genetics results and known protein-protein interactions in humans. By this, we have, in a data-driven fashion, narrowed down the list of proteins in the PPI network to the ones which have a higher likelihood of contributing to the disease through their corresponding genetic signatures and their mutual interactions. Then, we further overlay the identified 577 proteins to characterize if the abundance of any of these proteins are significantly impacted by PD SNPs. Therefore, during the process of hypothesis-free analysis, none of the 577 proteins is used and they are only used at the end to provide additional evidence and to narrow down the list of proteins in the top modules showing that the expression of these proteins is dysregulated by PD-related SNPs and at the same time their encoding genes are showing a high likelihood of disease association.

With respect to hypothesis-driven network analysis, we aimed to characterize if the expression of the interacting proteins with known PD proteins are being dysregulated because of PD genetic variants. This enables us to characterize proteins whose encoding genes may not harbor a genetic variant related to PD, but their expression can be impacted by such variants. Then, the dysregulation of these proteins can lead to interruption of normal protein-protein interactions. We showed that in fact this is the case on page 10 paragraph 2 reading “denoting significant enrichment of pQTL-associated proteins within the direct neighbors of PD GWAS loci in human PPI network”.

Finally, we came up with a short list of 14 proteins through merging the results of both network analyses. These proteins are the ones whose expression are significantly associated with PD genetic variants, their dysregulation leads to interruption of normal protein-protein interactions that can contribute to the disease, and are a member of top protein complexes (modules) which are significantly associated with the risk of the disease. In conclusion, we maintained that not all 577 proteins are disease-associated on page 6 paragraph 1. However, we need to consider them all for refinements of our findings and avoid any bias when making any conclusions. However, we acknowledge that, still, we have ~3000 proteins being profiled due to the existing technological limitations, and we hope that we can expand this work in the future once larger panels of proteins are available to the scientific community.

4. Reviewer's comment: *Further, the authors report that they restricted the GWAS sample size to “to establish the associations between genome using orthogonal data sets generated from the same samples and proteome”, but it is unclear the extent to which samples were orthogonal (I think there will be overlap?) and what benefit this brings? For example, it drastically reduces the extent of genetic signal for PD. And, the use of FinnGen data seem inconsistent with this statement. I don't understand the logic of the decision, especially because the analytical approaches used are two-sampled in nature. Perhaps the authors could qualify the additional insights gained through this approach?*

Author's Response: Thank you for raising this critical observation. By orthogonal, we meant that we have proteomics and genetics data on the same individuals in the UK Biobank (of course not all the participants in the UK Biobanks have proteomic data). This gives us the opportunity to directly study the genome-proteome interactions in the same set of patients. We totally agree with you that this may not be sufficient given that there are well powered genetics results on PD, such as the work by Nalls et al. Therefore, in this revision, we extended our work to use summary statistics provided by Nalls et al. and evaluated the associations between their reported SNPs and the proteins profiled in the UKB-PPP. We have explained our findings in page 7 paragraph 1 reading “The UKB-FinnGen meta-analysis is restricted in terms of the sample size. Therefore, we sought to leverage the summary statistics provided by Nalls et al. [1] and evaluate their potential associations with plasma protein concentrations in the UKB-PPP. 107 SNPs provided by Nalls et al. were evaluated. 51 SNPs were pQTLs in the UKB-PPP associated with a total of 204 proteins leading to a total number of 287 pQTL-protein associations (Supplementary Table 1). Out of 204 identified proteins, 149 proteins were shared with the 577 pQTL-associated proteins from UKB-FinnGen while 55 proteins were not identified before. pQTL-associated proteins are the proteins whose abundance is significantly associated with the SNPs identified in the UKB-FinnGen meta-analysis. We conducted a pathway enrichment analysis on the 204 proteins leading to a total of 6 significant pathways (Supplementary Table 3). Among which, three pathways were shared with the enriched pathways from the PD associated proteins from UKB-FinnGen including: lysosome (FDR=4.62E-4), Graft-versus-host disease (FDR=0.003), and Cell adhesion molecules (CAMs) (FDR=0.03). The other three pathways that are unique to this analysis include: Antigen processing and presentation (FDR=2.88E-5), Staphylococcus aureus infection (FDR=3.94E-4), and Natural killer cell mediated cytotoxicity (FDR=0.03)”.

In this revision, to complement our initial analysis, we conducted two-sample Mendelian Randomization using the summary statistics from Nalls et al. as the outcome, pQTL data from the UKB-PPP as the instrumental variables, and the protein abundances as the exposures to pinpoint putative causal proteins in PD. Moreover, we conducted a colocalization analysis to characterize if the UKB-FinnGen variants colocalize with the pQTLs associated with the proteins profiled in the UKB-PPP. Finally,

we merged these two and summarized all the MR results if any of the MR-derived proteins also show a colocalization signal. These results are discussed in page 8.

5. Reviewer's comment: *The authors variably refer to a wish to identify targets, which suggests that they wish to identify proteins with a causal link to disease – is this correct, and is this what is being suggested for the implicated proteins? Did the authors attempt to perform any form of causal analyses for these proteins? For example, for LGALS3 have other pQTLs and are they associated with PD? The authors cite previous MR analyses with what appear to be nominally significant findings – could the authors address the question using these expanded data?*

Author's Response: We appreciated your important comment. Indeed, our objective is to identify proteins with a causal link to disease, and this is suggested for the implicated proteins. We have conducted extensive causal inference analyses including MR and colocalization analysis (please see page 7; page 14 paragraph 1; Table 3; Supplementary 1). Conducting MR using the summary statistics from Nalls et al. and pQTLs from the UKB-PPP, revealed a significant causal association for LGALS3 in PD. Our finding matched the direction of association between LGALS3 and PD in the published MRs by Storm et al. and Kaiser et al. We outline this finding in page 13 paragraph 2 as follows “Through MR analysis, we found LGALS3 to significantly associate with PD (beta=0.22, FDR=0.001) indicating that its overexpression is putatively causal to increased risk of PD”.

6. Reviewer's comment: *The term “mediator” is also used throughout (alongside “indirect mediator”), which isn't particularly clear to this reader on what relationship they are suggested to mediate. Perhaps the authors could consider the use of directed graphs throughout to outline their hypotheses, as per ST-2.*

Author's Response: Thank you for raising this point. We have clarified the meaning of mediator in page 4 paragraph 1 as follows: “... but potential indirect mediators for which no sentinel genetic variant was identified while they may play critical roles in the course of the disease. Such mediators may either be proteins identified through causal inference analysis or through network analysis”.

7. Reviewer's comment: *Again, as I read the conclusions, some of the statements were unclear on what exactly they referred to. E.g. “Our study is the first effort to leverage proteogenomic UKB data in a network framework to characterize the genome-proteome relations and model their interactions.”*

Author's Response: Thank you for raising this point. To clarify and increase clarity, we rephrased this sentence as follows: “Our study is an unprecedented effort of integrating UKB proteogenomic data in a network framework to...”.

Reviewer 2: *In addition, there are a number of other more general or minor comments*

8. Reviewer's comment: *Generally, the text would benefit from being more concise – for example, the results section starts with over a page of additional exposition. Could this be shortened or interspersed with the relevant section of actual results?*

Author's Response: We have added a 'Study Design' section at the beginning of the 'Results' section to address your comment. In this section, details of the analytical pipeline as well as the data types used in each step are discussed. The outcomes of the study are later presented in the following sections under 'Results'.

9. Reviewer's comment: *Results, paragraph 2 – confirm which analyses were performed in step 1 – PD, I assume?*

Author's Response: Yes, it is PD. We changed the text to clarify your point.

10. Reviewer's comment: *Please provide citation for the pubic PPI network in the results when first mentioned.*

Author's Response: We have cited the data being used in page 4 paragraph 3 reading "... protein-protein interaction network (PPI) [29]".

11. Reviewer's comment: *"For the remaining nominated proteins, we did not observe published studies to demonstrate their links with neurodegenerative disorders." – the next sentence then reports a number of links for LGALS3.*

Author's Response: To make the text consistent, we added the following sentence to page 12 paragraph 3: "Several lines of evidence were found about LGALS3".

12. Reviewer's comment: *In the hypothesis-driven network analysis were the significant loci from the meta-analysis mapped to specific genes/proteins through the MAGMA gene-based scores? Same for the differential expression analyses of PD-associated loci with single-cell RNAseq data. It's unclear from the current text.*

Author's Response: In the hypothesis-driven analysis, we do not use gene-based scores. We extract the PD-associated nodes, identified from the UKB-FinnGen meta-analysis, along with their direct neighbors in the PPI network. For the DE analysis, we focus on genes which harbor a PD-associated genetic variant.

Reviewer 3

Author's Response: We would like to thank you for your deep and constructive comments which led to significant improvements to the manuscript.

Reviewer 4

This study performs an integrative analysis across complementary ‘omics datasets, that include recent proteomics results (UKB-PPP) that are anchored to genetic data from UKB to derive disease mechanisms and potential disease mediators of Parkinson’s disease (PD). Through these analyses the authors highlight PD-related pathways and proteins including LGALS3 for further study. I felt the paper was well written and showcases how population proteomics data can be used with orthogonal datasets to triangulate on genes with therapeutic potential in Parkinson's disease that might not be obvious from GWAS alone.

1. Reviewer’s comment: *It wasn’t clear to me what the benefit of conducting a separate PD GWAS compared to Nalls et al. was given the drop in case size and thus presumably power. The authors make a justification that their objective was to establish associations between the genome using orthogonal datasets generated from the same samples and proteome (page 5 line 126), however it was unclear what this overlap was, for example how many of the samples in the GWAS also had proteomic samples available in UKB?*

Author’s Response: Thank you so much for raising this important point. We agree with you about the limited sample size of our generated cohort in comparison to the cohort used by Nalls et al. Therefore, in the revised manuscript, we went ahead and used the summary statistics by Nalls et al. and conducted additional analyses to evaluate if the reported sentinel variants by Nalls et al. are pQTLs in the UKB-PPP data as well as their corresponding proteins followed by pathway enrichment analysis (Please see page 6 paragraph 4 and page 7 paragraph 1). Moreover, using Nalls et al. data and the pQTL results from the UKB-PPP, we performed a two-sample Mendelian Randomization analysis to characterize putative causal proteins in PD. We added a new section for this analysis in page 8 followed by summarizing the top findings in Table 3 as well as the detailed results in Supplementary Table 1. With respect to your comment about the number of samples with proteomic data, we added the following sentence to page 5: “Within our created case/control cohort used for running GWAS in the UKB, 638 PD patients and 15,522 healthy controls had proteomic data”.

Moreover, conducting hypothesis-free network analysis requires generating gene-based scores using all the available SNPs. The summary statistics provided by Nalls et al. only reports the top GWAS hits and does not provide genome-wide list of evaluated SNPs. Therefore, we are not able to run hypothesis-free network analysis only using results by Nalls et al. We have clarified this in page 5 paragraph 5 as follows: “Moreover, in order to generate gene-based scores to be used in the network analysis, meta-analysis was performed so that we later used the generated scores to weight the nodes of the PPIs.”.

2. Reviewer’s comment: *When integrating proteo-genomics results with PD it appears that a simple intersect was used (page 5 line 139), was there a reason*

that a more sophisticated approach such as colocalization (e.g. coloc) was not attempted? My concern is that with a simple intersect approach whilst a variant may be associated with both a protein's abundance and PD the underlying causal variant could be different which may generate unreliable results for downstream analyses. This information would be especially useful for the 'deep dive' on rs11158026-T and LGALS3.

Author's Response: Thank you for raising this critical point. Indeed, this is one of the major points that we have addressed in this revised version of the manuscript. We have now conducted a Mendelian Randomization (MR) and Colocalization analysis to characterize putative causal proteins associated with PD. These results have been added to page 8 reading "We conducted a Mendelian Randomization (MR) analysis to identify potential PD causally-linked proteins. We used 6,381 identified cis-pQTLs from the UKB-PPP as the instrumental variables and protein abundance as the Exposure in this analysis (Supplementary Table 1). We took advantage of summary statistics from Nalls et al. [37] to use in the MR analysis as the outcome (Methods and Materials). We identified a total of 122 proteins showing a significant association at FDR<0.05. Following MR, we conducted a colocalization analysis between the UKB-FinnGen meta-GWAS and pQTLs from the UKB-PPP to characterize if meta-analysis signals colocalize with pQTL hits in the MR-derived proteins aimed at assessing whether the genetic factors from meta-GWAS also colocalize with the genetic predictor of the protein abundances for supporting the MR findings which are based on the findings of Nalls et al. We found significant colocalization signals for 18 proteins identified in MR with a colocalization posterior probability >0.9. These 17 proteins are shown in Table 3 and the full list of all MR results is provided in Supplementary Table 1. In addition, we ran a colocalization analysis on the UKB-FinnGen meta-analysis genes (Table 2). Among which HIP1R (H4 posterior probability, H4.PP=1), GCH1 (H4.PP=0.99), SNCA (H4.PP=1), TMEM175 (H4.PP=0.95) show a significant colocalization signal".

Regarding LGALS3, MR results shows a significant association with PD which is in line with external MR results published in the literature. We outline this finding in page 13 paragraph 1 as follows "Through MR analysis, we found LGALS3 to significantly associate with PD (beta=0.22, FDR=0.001) indicating that its overexpression is putatively causal to increased risk of PD". In addition, the pQTL rs11158026 in *GCH1* associated with decreased expression of LGALS3 was found to be a significant colocalization signal. We added the following sentence in page 14 paragraph 1 reading "In addition, our colocalization analysis revealed rs11158026, a pQTL in *GCH1* that is significantly associated with expression of LGALS3, as a significant colocalization signal".

3. Reviewer's comment: *There are very sparse details on pathway enrichment method (page 6 line 151), for example was the fact that the selection of genes/proteins was limited to those assayed in UKB PPP considered? How were the dependencies between genes due to LD accounted for? Without these details hard to assess the relevance of this analysis.*

Author's Response: Thank you for your comment. Indeed, a major point when running a pathway enrichment analysis is to make sure to use a right reference set to avoid observing inflated p-values and superfluous pathways. In our study, to avoid finding false positive pathways, we used the list of proteins in the UKB-PPP as our reference set. We indicated this in page 15 as follows: "Pathway enrichment analysis was done using WebGestalt [83] using the Olink[®] protein panel used in the UKB-PPP as the reference". Of course, this panel is limited to ~3,000 proteins and does not cover the entire human proteome, which is a limitation of the existing technologies. As you rightly put, our analysis here is restricted to the proteins assayed in the UKB-PPP.

3. Reviewer's comment: *I wasn't entirely clear on the SNP-to-protein specificity section (page 6 line 163); how was LD considered etc. (especially given the Table2 contains HLA region). Is the observation that there are more trans associations than on the same chromosome surprising (page 6 line 175)? Given that the number of potential cis pQTL variants for a protein is an order of magnitude less than for trans/different chromosome pQTLs?*

Author's Response: Thank you for your comment. In this revision, we have revised the definition of cis associations (page 7 paragraph 3) to align it with the common definition for such associations in the literature reading "The frequencies of cis associations (defined as ± 1 kb flanking regions of gene coordinates) ...". As you rightly put, the number cis associations are smaller than the number of trans associations, so we removed the sentence in page 6 line 175 to avoid any misinterpretation. Our intention in this section is to only showcase the overall patterns of SNP-to-protein specificity and we have focused on only sentinel variants reported in Table 2. We do not consider SNPs that are in LD with these ten SNPs.

4. Reviewer's comment: *I thought hypothesis free PPI network analysis was fine as MAGMA was used to score genes which I believe makes some attempt to consider collinearity between GWAS stats due to LD. I found the hypothesis driven network (page 7 line 219) analysis harder to assess, given the strong assumptions (previously mentioned) that the closest gene is causal and therefore used to construct the network. For permutation analysis I wasn't convinced on the empirical distribution as the ~3K proteins assessed via UKB PPP are likely to be non-random in their selection, perhaps enriched for those with clinical application/association with disease, could that be an explanation for the enrichment?*

Author's Response: Thank you for your important input. In our study, we do not make a strong assumption that *the closest gene* in the network can necessarily be causal. However, given that at the protein level, they are directly interacting with PD GWAS hits-associated proteins, then their dysregulation can potentially lead to perturbations in normal functionality of PD proteins leading to potential contributions to the disease. This is our rationale to consider first-degree neighbors in the hypothesis-

driven network analysis. We acknowledge your comment about the proteins that have been selected by the UKB-PPP which are non-random. However, the protein panel is not biased for PD-related proteins. We totally agree with you that this is a limitation of our study. To clarify this to the readers, we have added the following statement in page 10 paragraph 2: “One of the limitations of this analysis is the constrained number of proteins profiled in the UKB-PPP. Therefore, hypothesis-driven analysis may not reflect the full spectrum of the human proteome and as a result, this might have led to overlooking some parts of the human interactome that might be involved in the disease”.

5. Reviewer’s comment: *For the chromatin loop analysis, which was interesting, I think fine mapping would have greatly increased the value of this analysis as this would provide some posterior estimate of a variant being causal. Whilst I understand the focus on brain tissue types, for context I think it worth looking across loop data for blood (given that proteomics was collected in plasma) to frame how brain specific the proposed regulatory variants are. This could bolster the point in the discussion on the relevance of plasma proteomics to neurodegenerative disease (somewhat undermined by focus on brain specific transcriptomic and epigenomic datasets – in my opinion).*

Author’s Response: Thank you for your comment. We ran a fine-mapping using the UKB-FinnGen GWAS results followed by mapping the significant hits, with $PP > 0.9$, onto the interactome map provided by Nott et al. [48] to show if any of the variants found to be falling in the identified regions in enhancer-promoter maps are putatively causal. Only two variants rs356219 (*SNCA*) and rs10847839 (*HIP1R*) were found to be significant hits in the fine-mapping process. We added the following paragraph to page 11 and also updated the Table 4 with the PP values of each SNP: “To complement the analysis, we conducted a fine-mapping (Methods and Materials) on the UKB-FinnGen meta-analysis results aimed at identifying the putative causal variants. Then we integrated them with the data by Nott et al. We consider a variant causal if the fine-mapping posterior probability (PP) > 0.9 . We found two variants rs356219 (*SNCA*) and rs10847839 (*HIP1R*) (Table 4) to be causal at $PP=1$ which were both identified to be acting in neuronal cells. These variants have also been reported by Schilder and Raj [52] to be causal in PD. The remaining variants reported in Table 4 did not have a $PP > 0.9$. Moreover, no other significant variants with $PP > 0.9$ were found to fall in the chromatin interactions loop provided by Nott et al. [53]”.

Regarding your second comment, we used blood-derived open chromatin data from healthy individuals to examine if any of the cell-specific PD variants identified through epigenomic analysis localize in open chromatin regions and can potentially exert regulatory impacts in blood. None of the PD variants were found to localize in such regions in blood implicating that the identified variants through epigenomic analysis, in fact act at specific cell-types in brain. We explained this investigation in page 11 paragraph 2 reading “Since the proteomic data in the UKB-PPP is plasma-derived, we sought to investigate if any of the identified cell-specific variants in Table 4 localize in

open chromatin regions in blood-derived data, to investigate if these variants are brain-specific. We used the assay for transposase-accessible chromatin using sequencing (ATAC-seq) data [52] processed by ATACdb [53] generated from healthy individuals. None of the brain-specific variants were found to be localizing in open chromatin regions in blood implicating that such variants have cell-specific regulatory roles in brain.”. We also noted this observation in the Discussion section.

6. Reviewer’s comment: *Page 11 line 345 does the association with rs11158026-T in PD and with LGALS3 colocalise such they are likely due to the same causal variant? I suspect that they do given the MR analysis referred to later on in this section but be good to have clarity.*

Author’s Response: Thank you for the delicate observation. Our MR analysis identifies LGALS3 as causal and this is shown in Table 3. Moreover, colocalization analysis shows rs11158026 as a significant hit (Page 14 paragraph 1).

7. Reviewer’s comment: *I was a bit baffled by the LGALS3 incident / prevalence association analyses with genotype (page 11 line 363) would not a Cox regression type approach be better suited to analysing incident cases to better understand the protective effect of rs11158026-T?*

Author’s Response: Our intention in page 11 was to dive deep into the genetic characteristics of different types of rs11158026 carriers and to interrogate how LGALS3 expression differs between such individuals. However, the pQTL analysis in fact considers incident/prevalent cases together and this observation is visualized in Fig. 7a. In addition, we did not intend to evaluate if this SNP affects the overall survival of patients, so we did not run a Cox model. Moreover, our observation on LGALS3 was further complemented by the causal inference analyses (MR and Coloc) that we conducted which basically provided a solid evidence about the potential causal association of LGALS3 with PD through rs11158026.

8. Reviewer’s comment: *The methods seem quite sparse as mentioned enrichment analysis is hard to interpret without a better understanding as to what was done. There are no methods for for single cell analysis (page 8 line 262) e.g. it says that significantly upregulated genes for each cluster were extracted but doesn’t say how.*

Author’s Response: We add additional description about the pathway enrichment analysis in page 17 paragraph 3 as follows “Pathway enrichment analysis was done using WebGestalt [84] using the Olink© protein panel used in the UKB-PPP as the reference. KEGG [37] biological pathways implemented in WebGestalt were used for enrichment analysis and pathways with FDR<0.05 were called significant”. With regard to page 8 line 262, we had directly used the summary statistics provided by Lake et al. [38] and had not processed the raw single cell data ourselves. We have now clarified

this in page 11 paragraph 3 reading “Using the summary statistics provided by Lake et al. [57], we extracted significantly...”.

9. Reviewer’s comment: *In terms of data availability – the GWAS summary statistics should be submitted to a public repository.*

Author’s Response: We have now submitted the generated summary statistics to GWAS Catalogue.

Minor comments

10. Reviewer’s comment: *Could the authors provide a qq plots/ an estimate for genomic control for their GWAS to provide reassurance that there is no confounding.*

Author’s Response: We have added the qq-plot to the supplementary figures and also indicated that in page 5 paragraph 3.

11. Reviewer’s comment: *For Table1 and Table2*

-can you add to legend what the listed allele is (I assume the effect allele) but good to make explicit.

Author’s Response: We modified the header of the first column to “SNP-effect allele” to make it explicit.

-could the gene column be renamed ‘Closest Gene’, also I think helpful to report MAF (for Finngen also for table2, so reader can assess founder effects etc.).

Author’s Response: We renamed the second column to ‘Closest gene’ in the both tables and included the MAF columns.

Perhaps worth mentioning that 12-40340400-G-A(rs34637584) given its rarity is also replicated using WES data (from Gene Bass).

Author’s Response: We mentioned this in page 5 paragraph 4 reading “Given that rs34637584 is a rare variant, we also checked Gene Bass [31] and we found that it is replicated through analyzing whole exome sequencing data (OR=3.37, P=3.81E-8).”.

Outside of LRRK2/APOE the associations appear to be non-coding common variants? Given there was no attempt to fine map and the fact that an index variant is not necessarily causal how relevant are V2G, CADD etc. columns?

Author’s Response: We have now conducted fine-mapping and added the posterior probabilities of these SNPs to Table 2.

12. Reviewer's comment: *Page 7 line 201 – what was the justification for top 1% was there an elbow in the scores or was this purely a pragmatic heuristic?*

Author's Response: We pragmatically used top 1%. We had tested the top 2% and 5% modules. However, we observed a large number of irrelevant genes to PD with small MAGMA scores. They were only shown up among the top module since they were making slight improvements in the objective function of the algorithm, due to the topology of the PPI network, while having no relevant biological relevance to PD.

13. Reviewer's comment: *Page7 line 204 what is the expectation of the number of shared proteins between the two groups i.e. does the observation of 14 proteins exceed that which we would expect by chance? I believe it does but good to have clarity.*

Author's Response: We tested if the overlap is significant. Using a hypergeometric test, the overlap was found to be significant at $P=0.003$. We added this to page 9 paragraph 3.

14. Reviewer's comment: *In suppl table 4 it would be helpful to add a column that details the genes within GWAS locus (red fig 5a) or with an intersecting pQTL.*

Author's Response: We add a column to this table and indicate your mentioned proteins.

15. Reviewer's comment: *Page7 line 210 abbreviation for AD is not previously defined suggest replacing with Alzheimer Disease.*

Author's Response: This is corrected.

16. Reviewer's comment: *Page 7 line 211-215 what is the genetic correlation between AD, PD and LBD? This might help a reader understand whether this degree of overlap is expected in terms of proteins/genes with putative shared aetiology.*

Author's Response: We added the following opening sentence to page 9 paragraph 4 to indicate shared genetic etiology between PD, AD, and LBD reading "Genetic correlations between PD, Alzheimer's disease, and Lewy body dementia (LBD) have been investigated in the literature [47, 48] and a few loci have been reported to be shared between these diseases such as *TMEM175*, *MAPT*, *KANSL1*".

17. Reviewer's comment: *Figure 1b – I see that chr23 is missing from the meta-analysis and there may be technical reasons for this. Given that there aren't any associations for chr23 it might be worth omitting from panel 'a' as well – in this way easier for reader to line up EHR GWAS with meta-analysis with FinnGen.*

Author's Response: We have now removed chr23 from Fig 2a as this chromosome was not available for FinnGen.

18. Reviewer's comment: *Figure 1c – I am not sure this adds much I would suggest removing – as there is no explanation of how p-values were calculated (tricky to do in the presence of LD).*

Author's Response: We have removed pane c in fig 2.

19. Reviewer's comment: *Fig 3a What are the units of scale on the x and y axis, seems to be chromosome? For example, is the line of trans associations near 5 on the x axis due to HLA association?*

Author's Response: We modified the axis titles to indicate that they represent the chromosomes.

20. Reviewer's comment: *Fig 3b As previously mentioned be great to see which of these actually colocalise rather than just intersect.*

Author's Response: The colocalization is now added to the Results section.

20. Reviewer's comment: *Figure 5a – consider using a different colors, red green not color-blind friendly also blue for pQTL associated proteins which are PD associated GWAS loci, is a bit confusing as default/background appears to be light blue..*

Author's Response: We corrected the background and re-created the panel (a), so the background is now white, to make sure the nodes are clearly visible. We also recreated the panel (b) per request of one of the reviewers to harmonize the colors between the two panels.

References

1. Wang, Q., Y. Liu, and J. Zhou, *Neuroinflammation in Parkinson's disease and its potential as therapeutic target*. *Transl Neurodegener*, 2015. **4**: p. 19.
2. Whitton, P.S., *Inflammation as a causative factor in the aetiology of Parkinson's disease*. *Br J Pharmacol*, 2007. **150**(8): p. 963-76.
3. Qu, Y., et al., *A systematic review and meta-analysis of inflammatory biomarkers in Parkinson's disease*. *NPJ Parkinsons Dis*, 2023. **9**(1): p. 18.
4. Witoelar, A., et al., *Genome-wide Pleiotropy Between Parkinson Disease and Autoimmune Diseases*. *JAMA Neurol*, 2017. **74**(7): p. 780-792.
5. Tansey, M.G., et al., *Inflammation and immune dysfunction in Parkinson disease*. *Nat Rev Immunol*, 2022. **22**(11): p. 657-673.

REVIEWER COMMENTS

Reviewer #1 (Remarks to the Author):

The authors did a great job improving the manuscript and addressing all comments. Congratulations.

I think the abbreviation "MR" is not explained the first time it is used, but other than that, I think the manuscript is ready.

Reviewer #2 (Remarks to the Author):

The authors have updated several sections of the paper in response to our comments, and the manuscript feels much improved. In particular, the inclusion of some attempt at causal inference is of real value. Below are a few specific comments for the authors to consider with relation to their updated manuscript.

To address the previous review comment that the stage 1 overlap of PD-associated SNPs and pQTLs may be confounded by LD, the authors performed MR analyses followed by colocalization. However, they still performed enrichment and network analyses using the proteins identified by the overlap of PD-associated SNPs and pQTLs. Because this overlap is likely to be confounded, any following analyses performed are also likely to be confounded. I might have expected that the colocalization-informed analyses would have served as the foundational analysis from which others were built, instead of just a follow-up analysis for just significant MR results. I realise that this would remove any proteins without pQTLs, so it may be that the balance is right, but I'd suggest some clarification on this approach and what it offers.

The MR analyses and results may benefit from additional clarification. MR was run using Nalls et al. as the outcome, but it seems that the colocalization analyses were performed with the UKB-FinnGen meta-analysis. Why was this choice made? It was my understanding that summary stats

are available from the Nalls et al GWAS – is that not correct? (if not, why not?) I also noted that the TwoSampleMR recommended $r^2 < 0.001$ threshold was not used, and was relaxed to $r^2 < 0.3$ instead. Was there a specific reason for this? Further, does the inclusion of Finnish data in the outcome GWAS bias towards the null in coloc analyses?

Generally, for the putatively causal proteins such as LGALS3, it would be of real interest to see a further deep dive into the MR results, the extent of pleiotropy in the instruments and heterogeneity in their causal estimates. I didn't note any investigation of instrument pleiotropy within the MR analyses. The SNP-to-protein specificity section suggests that it's possible that some of the instruments associate with hundreds of proteins and are highly pleiotropic.

The end of the paragraph describing the MR analyses and results states that colocalization analyses were also performed for UKB-FinnGen meta-analysis genes, and that four genes, including TMEM175, show colocalization. What signals were colocalized in this analysis? Was it Nalls et al. signals with UKB-FinnGen meta-analysis signals? If so, this should be stated. Currently, it reads as though these might be pQTL signals colocalizing with UKB-FinnGen meta-analysis signals. However, I didn't think that TMEM175 was measured on the Olink Explore panel.

As noted previously, it remains unclear to me what inference is to be drawn from the conclusions linking the 9 genes to PD pathogenesis in the abstract, or, even more so, for the 577 protein list. As noted by other reviewers, some of the enriched pathways are not intuitively relevant to PD. How should the reader interpret these insights and what are the next steps needed to translate these insights towards prevention or treatment of PD?

Reviewer #3 (Remarks to the Author):

I co-reviewed this manuscript with one of the reviewers who provided the listed reports as part of the Nature Communications initiative to facilitate training in peer review and appropriate recognition for co-reviewers

Reviewer #4 (Remarks to the Author):

Thanks for addressing my comments – there are a couple of follow up points that I think need addressing before publication that I think will strengthen the work.

1. Reviewer’s comment: It wasn’t clear to me what the benefit of conducting a separate PD GWAS compared to Nalls et al. was given the drop in case size and thus presumably power. The authors make a justification that their objective was to establish associations between the genome using orthogonal datasets generated from the same samples and proteome (page 5 line 126), however it was unclear what this overlap was, for example how many of the samples in the GWAS also had proteomic samples available in UKB?

Author’s Response: Thank you so much for raising this important point. We agree with you about the limited sample size of our generated cohort in comparison to the cohort used by Nalls et al. Therefore, in the revised manuscript, we went ahead and used the summary statistics by Nalls et al. and conducted additional analyses to evaluate if the reported sentinel variants by Nalls et al. are pQTLs in the UKB-PPP data as well as their corresponding proteins followed by pathway enrichment analysis (Please see page 6 paragraph 4 and page 7 paragraph 1). Moreover, using Nalls et al. data and the pQTL results from the UKB-PPP, we performed a two-sample Mendelian Randomization analysis to characterize putative causal proteins in PD. We added a new section for this analysis in page 8 followed by summarizing the top findings in Table 3 as well as the detailed results in Supplementary Table 1. With respect to your comment about the number of samples with proteomic data, we added the following sentence to page 5: “Within our created case/control cohort used for running GWAS in the UKB, 638 PD patients and 15,522 healthy controls had proteomic data”.

Moreover, conducting hypothesis-free network analysis requires generating gene-based scores using all the available SNPs. The summary statistics provided by Nalls et al. only reports the top GWAS hits and does not provide genome-wide list of evaluated SNPs.

Is this true that summary results for Nalls et al. (with 23&me removed perhaps) are available at http://ftp.ebi.ac.uk/pub/databases/gwas/summary_statistics/GCST009001-GCST010000/GCST009325/ ? Are these not of use/relevance to your analyses.

Therefore, we are not able to run hypothesis-free network analysis only using results by Nalls et al. We have clarified this in page 5 paragraph 5 as follows: “Moreover, in order to generate gene-based scores to be used in the network analysis, meta-analysis was performed so that we later used the generated scores to weight the nodes of the PPIs.”

See above comment

2. Reviewer’s comment: When integrating proteo-genomics results with PD it appears that a simple intersect was used (page 5 line 139), was there a reason that a more sophisticated approach such as colocalization (e.g. coloc) was not attempted? My concern is that with a simple intersect approach whilst a variant may be associated with both a protein’s abundance and PD the underlying causal variant could be different which may generate unreliable results for downstream analyses. This information would be especially useful for the ‘deep dive’ on rs11158026-T and LGALS3.

Author’s Response: Thank you for raising this critical point. Indeed, this is one of the major points that we have addressed in this revised version of the manuscript. We have now conducted a Mendelian Randomization (MR) and Colocalization analysis to characterize putative causal proteins associated with PD. These results have been added to page 8 reading “We conducted a Mendelian Randomization (MR) analysis to identify potential PD causally-linked proteins. We used 6,381 identified cis-pQTLs from the UKB-PPP as the instrumental variables and protein abundance as the Exposure in this analysis (Supplementary Table 1). We took advantage of summary statistics from Nalls et al. [37] to use in the MR analysis as the outcome (Methods and Materials). We identified a total of 122 proteins showing a significant association at $FDR < 0.05$. Following MR, we conducted a colocalization analysis between the UKB-FinnGen meta-GWAS and pQTLs from the UKB-PPP to characterize if meta-analysis signals colocalize with pQTL hits in the MR-derived proteins aimed at assessing whether the genetic factors from meta-GWAS also colocalize with the genetic predictor of the protein abundances for supporting the MR findings which are based on the findings of Nalls et al. We found significant colocalization signals for 18 proteins identified in MR with a colocalization posterior probability > 0.9 . These 17 proteins are shown in Table 3 and the full list of all MR results is provided in Supplementary Table 1. In addition, we ran a colocalization analysis on the UKB-FinnGen meta-analysis genes (Table 2). Among which HIP1R (H4 posterior probability, H4.PP=1), GCH1 (H4.PP=0.99), SNCA (H4.PP=1), TMEM175 (H4.PP=0.95) show a significant colocalization signal”.

In supplementary table 1 does the nsnp column indicate the number of SNPs used in the MR (which I suspect), could perhaps another column be added to indicate the number of SNPs used for colocalization (this is in the output of coloc)?

“Following MR, we conducted a colocalization analysis between the UKB-FinnGen meta-GWAS and pQTLs from the UKB-PPP to characterize if meta-analysis signals colocalize with pQTL hits in the MR-derived proteins aimed at assessing whether the genetic factors from meta-GWAS also colocalize with the genetic predictor of the protein abundances for supporting the MR findings which are based on the findings of Nalls et al.” – Can this be rewritten to be clearer – it is a very long sentence that I found hard to parse.

In the sentence beginning “We found significant colocalization signals for 18 proteins identified” consider replacing with “We found strong colocalization support (H4.PP>0.9) for a shared causal variant for abundance and PD risk for 18 proteins identified through MR” – significant implies a frequentist approach was used which I don’t think was the case.

Why only 17 proteins shown in Table 3 – is this a typo should it be 18?

“In addition, we ran a colocalization analysis on the UKB-FinnGen meta-analysis genes (Table 2)” I assume colocalization between UKB-PPP pQTL summary stats and UKB-FinnGen meta-analysis – please make this clear if so. Also replace “genes” with loci

In the sentence beginning “Among which HIP1R” – please replace significant with ‘strong support for colocalization’

3. Reviewer’s comment: There are very sparse details on pathway enrichment method (page 6 line 151), for example was the fact that the selection of genes/proteins was limited to those assayed in UKB PPP considered? How were the dependencies between genes due to LD accounted for? Without these details hard to assess the relevance of this analysis.

Author’s Response: Thank you for your comment. Indeed, a major point when running a pathway enrichment analysis is to make sure to use a right reference set to avoid observing inflated p-values and superfluous pathways. In our study, to avoid finding false positive pathways, we used the list of proteins in the UKB-PPP as our reference set. We indicated this in page 15 as follows: “Pathway enrichment analysis was done using WebGestalt [83] using the Olink© protein panel used in the UKB-PPP as the reference”. Of course, this panel is limited to ~3,000 proteins and does not cover the entire human proteome, which is a limitation of the existing technologies. As you rightly put, our analysis here is restricted to the proteins assayed in the UKB-PPP.

Please can you explain how you accounted for confounding because of, for example LD, in these pathway enrichment analyses? Whilst associated genes may be independent (as I believe you

selected one gene/protein from a region) from each other this may not be the case in the null set. Why not use a tool specifically developed for these kinds of analyses for example MAGMA (as used later)?

3. Reviewer's comment: I wasn't entirely clear on the SNP-to-protein specificity section (page 6 line 163); how was LD considered etc. (especially given the Table2 contains HLA region). Is the observation that there are more trans associations than on the same chromosome surprising (page 6 line 175)? Given that the number of potential cis pQTL variants for a protein is an order of magnitude less than for trans/different chromosome pQTLs?.

Author's Response: Thank you for your comment. In this revision, we have revised the definition of cis associations (page 7 paragraph 3) to align it with the common definition for such associations in the literature reading "The frequencies of cis associations (defined as ± 1 kb flanking regions of gene coordinates) ...". As you rightly put, the number cis associations are smaller than the number of trans associations, so we removed the sentence in page 6 line 175 to avoid any misinterpretation. Our intention in this section is to only showcase the overall patterns of SNP-to-protein specificity and we have focused on only sentinel variants reported in Table 2. We do not consider SNPs that are in LD with these ten SNPs.

Perhaps a misunderstanding but why not use proteins that colocalise with signals from Table2 for this analysis now rather than pQTLs that intersect with these PD associated variants?

5. Reviewer's comment: For the chromatin loop analysis, which was interesting, I think fine mapping would have greatly increased the value of this analysis as this would provide some posterior estimate of a variant being causal. Whilst I understand the focus on brain tissue types, for context I think it worth looking across loop data for blood (given that proteomics was collected in plasma) to frame how brain specific the proposed regulatory variants are. This could bolster the point in the discussion on the relevance of plasma proteomics to neurodegenerative disease (somewhat undermined by focus on brain specific transcriptomic and epigenomic datasets – in my opinion).

Author's Response: Thank you for your comment. We ran a fine-mapping using the UKB-FinnGen GWAS results followed by mapping the significant hits, with $PP > 0.9$, onto the interactome map provided by Nott et al. [48] to show if any of the variants found to be falling in the identified regions in enhancer-promoter maps are putatively causal. Only two variants rs356219 (SNCA) and rs10847839 (HIP1R) were found to be significant hits in the fine-mapping process. We added the following paragraph to page 11 and also updated the Table 4 with the PP values of each SNP: "To complement the analysis, we conducted a fine-mapping (Methods and Materials) on the UKB-FinnGen meta-analysis results aimed at identifying the putative causal variants. Then we integrated them with the data by Nott et al. We consider a variant causal if the fine-mapping posterior probability (PP) > 0.9 . We found two variants rs356219 (SNCA) and rs10847839 (HIP1R) (Table 4) to be causal at $PP=1$ which were both identified to be acting in neuronal cells. These variants have

also been reported by Schilder and Raj [52] to be causal in PD. The remaining variants reported in Table 4 did not have a $PP > 0.9$. Moreover, no other significant variants with $PP > 0.9$ were found to fall in the chromatin interactions loop provided by Nott et al. [53]”.

Regarding your second comment, we used blood-derived open chromatin data from healthy individuals to examine if any of the cell-specific PD variants identified through epigenomic analysis localize in open chromatin regions and can potentially exert regulatory impacts in blood. None of the PD variants were found to localize in such regions in blood implicating that the identified variants through epigenomic analysis, in fact act at specific cell-types in brain. We explained this investigation in page 11 paragraph 2 reading “Since the proteomic data in the UKB-PPP is plasma-derived, we sought to investigate if any of the identified cell-specific variants in Table 4 localize in open chromatin regions in blood-derived data, to investigate if these variants are brain-specific. We used the assay for transposase-accessible chromatin using sequencing (ATAC-seq) data [52] processed by ATACdb [53] generated from healthy individuals. None of the brain-specific variants were found to be localizing in open chromatin regions in blood implicating that such variants have cell-specific regulatory roles in brain.”. We also noted this observation in the Discussion section.

Thanks for adding the SuSIE fine-mapping. How did you define loci/window around each signal in which to perform the fine-mapping? I assume you used a reference to compute the required LD matrix given that this was on a meta-analysis? Can you add some details on that in methods, as this will help readers to interpret how this might affect finemapping results depending on how reference accurately models LD differences between FinnGen and UKB. Can the 95% credible sets for each locus be added as a supplementary table? Did some loci appear to have more than 1 credible set?

“We consider a variant causal if the fine-mapping posterior probability (PP) > 0.9 ” What was the reason for $P(I)P > 0.9$ it seems very stringent? some papers such as [PMID: 33828297] use > 0.1 , perhaps you can add a supporting reference or some justification as to why you decided on > 0.9 ?

For MR analyses what filters/follow up analyses did you put in place to examine whether the results might be affected by violation of MR assumptions (e.g. horizontal pleiotropy)?

9. Reviewer’s comment: In terms of data availability – the GWAS summary statistics should be submitted to a public repository.

Author’s Response: We have now submitted the generated summary statistics to GWAS Catalogue.

Thanks for this – do you have an accession (even if this is embargoed) as this would be useful to refer to in the paper if possible?

Minor comments

There seems to be some issues with citations please check and update these.

Consider changing the title of Table4 to The list of the PD variants overlapping (rather than enriched) regulatory regions

Responses to Reviewers' Comments

Summary of changes

We would like to thank the editor and the reviewers for your evaluation of our manuscript (NCOMMS-23-34002A) titled 'Proteogenomic network analysis reveals dysregulated mechanisms and potential mediators in Parkinson's disease'. We have carefully addressed all of the comments raised by the editor and the reviewers and made modifications to the text. Please find below our point-by-point responses to the reviewers' specific comments. All page numbers refer to the revised manuscript file, and relevant sections have been yellow highlighted.

Reviewer 1

The authors did a great job improving the manuscript and addressing all comments. Congratulations.

I think the abbreviation "MR" is not explained the first time it is used, but other than that, I think the manuscript is ready.

Author's Response: We would like to thank you for your constructive and insightful comments. Your feedback has led to significant improvements to the manuscript. To your point, we added "MR" where it was first used.

Reviewer 2

The authors have updated several sections of the paper in response to our comments, and the manuscript feels much improved. In particular, the inclusion of some attempt at causal inference is of real value. Below are a few specific comments for the authors to consider with relation to their updated manuscript.

1. Reviewer's comment: *To address the previous review comment that the stage 1 overlap of PD-associated SNPs and pQTLs may be confounded by LD, the authors performed MR analyses followed by colocalization. However, they still performed enrichment and network analyses using the proteins identified by the overlap of PD-associated SNPs and pQTLs. Because this overlap is likely to be confounded, any following analyses performed are also likely to be confounded. I might have expected that the colocalization-informed analyses would have served as the foundational analysis from which others were built, instead of just a follow-up analysis for just significant MR results. I realise that this would remove any proteins without pQTLs, so it may be that the balance is right, but I'd suggest some clarification on this approach and what it offers.*

Author's Response: Thank you for raising this important point. In general, we ran two sets of network analyses: hypothesis-free and hypothesis-driven. The goal of hypothesis-free analysis is to leverage the UKB-FinnGen meta-analysis results, mapping them to their corresponding nodes in the PPI network, conducting a network search, and characterizing a set of protein complexes (modules) that show the highest association with the disease risk. Please note that the process of mapping the summary statistics onto the nodes in the network, encompasses all the SNPs within the boundaries of the corresponding node (gene) in the PPI network. As a result, the potential LD is already accounted for in MAGMA when we obtain the weight of each node. This process is completely independent of the pQTLs derived from the UKB-PPP and the main objective is to find genetically-driven sets of proteins whose aggregated interactions can potentially be associated with the disease risk. Next, for further exploration we sought to check if any of the proteins that fall within the identified significant modules are associated with independent PD SNPs which are also pQTLs (as shown in green in Fig 5a). The same applies to the hypothesis-driven network analysis. In this analysis, nodes in the PPI network that harbor an independent significant PD-associated SNP were extracted along with their directly interacting proteins. Here, our objective is to investigate if the abundance of proteins, that directly interact with PD proteins, is associated with the significant PD variants related to that PD protein. We acknowledge that some proteins that are differentially expressed might not show up here, however that is not the scope of this analysis. Therefore, the network analysis results are independent of LD confounding effects that you indicated. We hope this clarifies our work.

2. Reviewer's comment: *The MR analyses and results may benefit from additional clarification. MR was run using Nalls et al. as the outcome, but it seems that the colocalization analyses were performed with the UKB-FinnGen meta-analysis. Why was this choice made? It was my understanding that summary stats are available from the Nalls et al GWAS – is that not correct? (if not, why not?) I also noted that the TwoSampleMR recommended $r^2 < 0.001$ threshold was not used, and was relaxed to $r^2 < 0.3$ instead. Was there a specific reason for this? Further, does the inclusion of Finnish data in the outcome GWAS bias towards the null in coloc analyses?*

Author's Response: Thank you for your comment. With respect to your point with r^2 , our analysis was based on the recommended $r^2 < 0.001$ and we have amended the typo ($r^2 < 0.3$ was a separate analysis that is not included in the manuscript). Regarding to your point on colocalization, we ran another analysis between the UKB-FinnGen and the summary statistics from Nalls et al. We added the details to page 7 paragraph 3 as follows: “We performed another colocalization analysis between the pQTLs from the UKB-PPP and GWAS results by Nalls et al. [1]. We found three loci BST1 (PP.H4=0.95), GPNMB (PP.H4=0.94) and CTSB (PP.H4=0.96) with showing a colocalization support (H4.PP>0.9). Among these loci, BST1 was found as a colocalization support in the analysis indicated above, while the other two genes were not identified as colocalization signals in the analysis noted above. These three genes were also found as significant MR evidences in our analysis”

Regarding your point on including the FinnGen data, we note that despite the improvement of the meta-analysis UKB-FinnGen as opposed to UKB GWAS, we confirm that signals obtained from the UKB cohort are more significant than the associations reported by FinnGen, where only two significant loci is found in PD. As a result, we do not believe if meta analyzing UKB with FinnGen GWAS creates a bias toward null in the colocalization analysis.

3. Reviewer's comment: *Generally, for the putatively causal proteins such as LGALS3, it would be of real interest to see a further deep dive into the MR results, the extent of pleiotropy in the instruments and heterogeneity in their causal estimates. I didn't note any investigation of instrument pleiotropy within the MR analyses. The SNP-to-protein specificity section suggests that it's possible that some of the instruments associate with hundreds of proteins and are highly pleiotropic.*

Author's Response: Thank you for raising this important point. Upon running MR on cis-pQTLs, we conducted additional analysis to test the horizontal pleiotropy. No significant horizontal pleiotropy was observed. We have indicated this in page 7. To your point, we further investigated the instrumental variable associated with LGALS3 and observed no significant pleiotropic association it. We have added this to page 14 paragraph 2.

4. Reviewer's comment: *The end of the paragraph describing the MR analyses and results states that colocalization analyses were also performed for UKB-FinnGen meta-analysis genes, and that four genes, including TMEM175, show colocalization. What signals were colocalized in this analysis? Was it Nalls et al. signals with UKB-FinnGen meta-analysis signals? If so, this should be stated. Currently, it reads as though these might be pQTL signals colocalizing with UKB-FinnGen meta-analysis signals. However, I didn't think that TMEM175 was measured on the Olink Explore panel.*

Author's Response: Thank you for noticing this error. We have modified the text stating that these are colocalized signals with Nalls et al.

4. Reviewer's comment: *As noted previously, it remains unclear to me what inference is to be drawn from the conclusions linking the 9 genes to PD pathogenesis in the abstract, or, even more so, for the 577 protein list. As noted by other reviewers, some of the enriched pathways are not intuitively relevant to PD. How should the reader interpret these insights and what are the next steps needed to translate these insights towards prevention or treatment of PD?*

Author's Response: Thank you for your comment. The primary objective of this work has been to identify the proteins that are affected by the PD-associated variants. Proteins are the pillars of cellular machinery, and they are considered as therapeutic targets. Therefore, characterizing the impact of genetic variations on proteomic disruptions can give us a better understanding about the affected biological pathways that may give rise to the disease risk. Regarding your other point about the identified pathways in the previous revision, we clarified this to the reviewer reading "The data generated by the UKB-PPP is not dedicated to PD and the list of proteins in the Olink panel include low-abundant inflammatory proteins, proteins actively secreted into blood circulation, approved and ongoing drug targets, organ-specific proteins leaked into blood circulation, and the proteins representing exploratory biomarkers. Therefore, not all the proteins implicated in PD have been profiled in this consortium. On the other hand, neuroinflammation plays a critical role in the pathogenesis of PD. In fact, central and peripheral inflammation play a vital role in pathological features of PD [1, 2]. Among inflammatory markers, chemokines such as chemokine ligand (CCL) and CXCL family as well as cytokines such as tumour necrosis factor (TNF) and interleukin (IL) are known as critical signaling molecules of immune activation in the central nervous system [3]. Looking into the enriched proteins provided in Supplementary Table 3, we can see that the top pathways are predominantly enriched by immune-related markers such as CCLs, ILs, and CXCLs. This is in line with our existing knowledge about the role of neuroinflammation in the PD pathogenesis. Moreover, PD is known to share several GWAS loci with multiple autoimmune disorders such as rheumatoid arthritis [4, 5]. This explains observing immune related pathways such as 'Cytokine-cytokine receptor interaction', 'Rheumatoid arthritis', 'Chemokine signaling pathway', and 'Intestinal immune network for IgA production'. Taken together with the fact the UKB-PPP is not a PD-dedicated study which is designed to reflect the entire population in the

UK Biobank, not all the PD-related genes and their encoded proteins have been profiled by the UKB-PPP. Therefore, we totally agree with your delicate point and necessary clarifications have been made to the text". Therefore, with this work, we lay the groundwork for the broader audience in the scientific community to gain a big picture of all the genome-proteome interactions occurring in PD for further investigations such as novel target discovery or biomarker identification. To reflect on your point, we added the following to the end of the Discussion section reading "This work lays the groundwork to gain a bigger picture about the reflection of genetic aberrations on the human proteome and that how such affected proteins can be further studied and validated for future purposes such as novel target discovery and biomarker identification".

Reviewer 3

I co-reviewed this manuscript with one of the reviewers who provided the listed reports as part of the Nature Communications initiative to facilitate training in peer review and appropriate recognition for co-reviewers.

Author's Response: We would like to thank you for your deep and constructive comments which led to significant improvements to the manuscript.

Reviewer 4

Thanks for addressing my comments – there are a couple of follow up points that I think need addressing before publication that I think will strengthen the work.

1. Reviewer's comment: *Is this true that summary results for Nalls et al. (with 23&me removed perhaps) are available at http://ftp.ebi.ac.uk/pub/databases/gwas/summary_statistics/GCST009001-GCST010000/GCST009325/ ? Are these not of use/relevance to your analyses.*

Therefore, we are not able to run hypothesis-free network analysis only using results by Nalls et al. We have clarified this in page 5 paragraph 5 as follows: "Moreover, in order to generate gene-based scores to be used in the network analysis, meta-analysis was performed so that we later used the generated scores to weight the nodes of the PPIs."

See above comment

Author's Response: Thank you so much for sharing the link with us. We went ahead, downloaded the data from the link you kindly shared and ran a screening check. We can see significant discrepancies between the statistics reported in this data with the top hits reported in the Supplementary Table 2 by Nalls et al. To name a few, in the supplementary file in the Nall's paper the beta=0.1351, however, the beta value reported in this file is -0.1127. In another example, rs9912362 is reported with a beta 0.077 in the supplementary table of Nall's paper. However, the reported beta for this SNP in this file is -0.037. Therefore, due to these inconsistencies, we believe this data is not suitable to be used in the network analysis. With that, we managed to obtain the summary statistics of Nalls's paper and ran a second colocalization analysis with the pQTL data from the UKB-PPP. The results have been indicated in page 7 paragraph 3.

2. Reviewer's comment: *In supplementary table 1 does the nsnps column indicate the number of SNPs used in the MR (which I suspect), could perhaps another column be added to indicate the number of SNPs used for colocalization (this is in the output of coloc)?*

Author's Response: That is correct. In Supplementary Table 1, nsnps indicates the number of SNPs (i.e., instrumental variables). We have now added another column to indicate the number of SNPs that had led to significant colocalization signal. For the non-significant signals, coloc does not output number of SNPs.

3. Reviewer's comment: *In the sentence beginning "We found significant colocalization signals for 18 proteins identified" consider replacing with "We found strong colocalization support ($H4.PP > 0.9$) for a shared causal variant for abundance and PD risk for 18 proteins identified through MR" – significant implies a frequentist approach was used which I don't think was the case.*

Why only 17 proteins shown in Table 3 – is this a typo should it be 18?

Author's Response: Thank you for your suggestion. We have now replaced it with the new sentence. We also noticed the type and corrected it.

4. Reviewer's comment: *Please can you explain how you accounted for confounding because of, for example LD, in these pathway enrichment analyses? Whilst associated genes may be independent (as I believe you selected one gene/protein from a region) from each other this may not be the case in the null set. Why not use a tool specifically developed for these kinds of analyses for example MAGMA (as used later)?*

Author's Response: Thank for raising this point. Based on your recommendation we ran MAGMA for pathway enrichment analysis and added the findings to page 7 paragraph 1 reading: "Moreover, we conducted an enrichment analysis on the SNPs that were found to be pQTLs in the UKB-PPP using MAGMA [32] to account for potential confounding factors caused by linkage disequilibrium (LD). We observed a few enriched pathways including Cytokine-cytokine receptor interaction (FDR = 1.7E-5), Chemokine signaling pathway (FDR=2.7E-7), Lysosome (FDR=1.5E-7), and Rheumatoid arthritis (FDR=2.3E-5). These observations are in accordance with our pathway enrichment results applied to the pQTL-associated proteins".

5. Reviewer's comment: *Perhaps a misunderstanding but why not use proteins that colocalise with signals from Table2 for this analysis now rather than pQTLs that intersect with these PD associated variants?*

Author's Response: Thank for providing additional clarification. We made additional investigation on the sentinel variants reported in Table 2. We should note that not all the genes reported in Table 2 are profiled in the UKB-PPP and only GBA, SNCA, HIP1R, and NECTIN2 are available in the UKB-PPP data. The following paragraph is added to page 8 reading "We investigated the specificity of the SNPs reported in Table 2 to proteins profiled in the UKB-PPP. rs2230288 (missense variant in GBA) shows one significant association in cis with EFNA1. rs34311866 (missense variant in TMEM175) was observed to significantly associate with IDUA in cis (Beta=-0.05, P=4.6E-8). Of note, rs2760980 (an intergenic variant close to HLA-DRB1) shows 98 significant associations among which five associations were in cis while the rest were in trans. The cis-associated proteins included AGER, AIF1, DXO, HLA-DRA, and TNXB. We found that rs10847839 (an intronic variant in HIP1R) is in fact associated with HIP1R in the UKB-PPP (Beta=0.09, P=6.7E-30). rs11158026 (intronic to GCH1) was significantly associated with decreased levels of LGALS3 (Beta=-0.28, -Log₁₀(P)=316). rs4630591 (intronic to KANSL1) shows 21 significant association among which only one association is in cis with LRRC37A2 (Beta=1.07, -Log₁₀(P)=2751) while the remaining association show -Log₁₀(P)<13. Finally, rs6857 (NECTIN2) was found to be highly

pleiotropic with 50 significant associations among which only one association in cis with APOE was identified (Beta=-0.85, -Log10(P)=1573). We can see that these variants are mostly specific to significant cis associations and a few of them are highly pleiotropic with larger p-values in trans associations compared to cis associations.”

6. Reviewer’s comment: *Thanks for adding the SuSIE fine-mapping. How did you define loci/window around each signal in which to perform the fine-mapping? I assume you used a reference to compute the required LD matrix given that this was on a meta-analysis? Can you add some details on that in methods, as this will help readers to interpret how this might affect finemapping results depending on how reference accurately models LD differences between FinnGen and UKB. Can the 95% credible sets for each locus be added as a supplementary table? Did some loci appear to have more than 1 credible set?*

Author’s Response: For fine-mapping, we used a 500kb window flanking independent SNPs. To create LD matrix, 1000 Genomes phase 3 reference panel was used. To your point, we added a new tab in Supplementary Table 1 and added the 95% credible sets for each significant locus; we did not observe loci to have more than one credible set. We have also expanded the fine-mapping part in the Methods section in page 19.

7. Reviewer’s comment: *For MR analyses what filters/follow up analyses did you put in place to examine whether the results might be affected by violation of MR assumptions (e.g. horizontal pleiotropy)?*

Author’s Response: We had tested for horizontal pleiotropy using MRPRESSO v1.0 R package. None of the MR results showed significant horizontal pleiotropy. We have updated the MR description in the Methods section.

8. Reviewer’s comment: *Thanks for this – do you have an accession (even if this is embargoed) as this would be useful to refer to in the paper if possible?*

Author’s Response: Absolutely. The accession number is GCST90319903. We have also indicated this in the Methods section.

9. Reviewer’s comment: *There seems to be some issues with citations please check and update these.*

Author’s Response: We double checked the citations and ensured all are correct.

10. Reviewer’s comment: *Consider changing the title of Table4 to The list of the PD variants overlapping (rather than enriched) regulatory regions*

Author’s Response: We have now modified the title of Table 4.

References

1. Wang, Q., Y. Liu, and J. Zhou, Neuroinflammation in Parkinson's disease and its potential as therapeutic target. *Transl Neurodegener*, 2015. 4: p. 19.
2. Whitton, P.S., Inflammation as a causative factor in the aetiology of Parkinson's disease. *Br J Pharmacol*, 2007. 150(8): p. 963-76.
3. Qu, Y., et al., A systematic review and meta-analysis of inflammatory biomarkers in Parkinson's disease. *NPJ Parkinsons Dis*, 2023. 9(1): p. 18.
4. Witoelar, A., et al., Genome-wide Pleiotropy Between Parkinson Disease and Autoimmune Diseases. *JAMA Neurol*, 2017. 74(7): p. 780-792.
5. Tansey, M.G., et al., Inflammation and immune dysfunction in Parkinson disease. *Nat Rev Immunol*, 2022. 22(11): p. 657-673.

REVIEWER COMMENTS

Reviewer #2 (Remarks to the Author):

The authors corrected several of the points we raised. Below, some suggestions that could be considered for clarity.

I'd suggest including the actual results from the sensitivity analyses applied to the MR results in a supplementary table or figure, and describing the key results and their relevance (or lack of) could be helpful. For example, for LGALS3, the text mentions that there is a significant coloc, but Supplementary Table 1 (which I assume is "MR-Coloc" in the excel files in my zip file, but it doesn't explicitly label it) lists the coloc H4 posterior is 0.1. Was there another analysis that showed a different result?

The inclusion of coloc results from UKB and the Nall results is useful. I did find the inserted text a little hard to follow (line 229-236). Are the results of these colocs shown in Supplementary Table 1? A clearer description of how these two different coloc results were integrated may be helpful to the reader. For example, was one considered higher value than the other, or was a positive result from either deemed worthy, despite a negative result in the other?

Also, a note of clarification on my earlier question: my question on bias towards the null using FinnGen was related more to the different genetic architecture than the sample size. As above, describing those two sets of results and their overlap and distinct features will address that empirically.

Reviewer #3 (Remarks to the Author):

Reviewer #4 (Remarks to the Author):

I thank the authors for answering my questions, I have no further concerns with regards to publication.

Responses to Reviewers' Comments

Summary of changes

We would like to thank the editor and the reviewers for your evaluation of our manuscript (NCOMMS-23-34002B) titled 'Proteogenomic network analysis reveals dysregulated mechanisms and potential mediators in Parkinson's disease'. We have carefully addressed all of the comments raised by the editor and the reviewers and made modifications to the text. Please find below our point-by-point responses to the reviewers' specific comments. All page numbers refer to the revised manuscript file, and relevant sections have been yellow highlighted.

Reviewer 2

The authors corrected several of the points we raised. Below, some suggestions that could be considered for clarity.

Reviewer's comment: I'd suggest including the actual results from the sensitivity analyses applied to the MR results in a supplementary table or figure, and describing the key results and their relevance (or lack of) could be helpful. For example, for LGALS3, the text mentions that there is a significant coloc, but Supplementary Table 1 (which I assume is "MR-Coloc" in the excel files in my zip file, but it doesn't explicitly label it) lists the coloc H4 posterior is 0.1. Was there another analysis that showed a different result?

Author's Response: Thank you for raising this point. We have added a new column to the MR-Coloc tab in Supplementary Table 1 and included MR leave-one-out sensitivity analysis results there. Regarding to your point on LGALS3, in fact the point that was made in the text was based on another analysis, which we are not confident about, so we have removed the sentence from paragraph 1 page 15.

Reviewer's comment: The inclusion of coloc results from UKB and the Nall results is useful. I did find the inserted text a little hard to follow (line 229-236). Are the results of these colocs shown in Supplementary Table 1? A clearer description of how these two different coloc results were integrated may be helpful to the reader. For example, was one considered higher value than the other, or was a positive result from either deemed worthy, despite a negative result in the other?

Also, a note of clarification on my earlier question: my question on bias towards the null using FinnGen was related more to the different genetic architecture than the sample

size. As above, describing those two sets of results and their overlap and distinct features will address that empirically.

Author's Response: The MR and colocalization results that are included in Supplementary Table 1 outline the colocalization analysis between the UKB-FinnGen meta-analysis and pQTLs from the UKB-PPP. For the other two coloc results, i.e. UKB-FinnGen and Nalls, and UKB-PPP and Nalls, we have not included them in the supplementary table and have outlined them in the text. We have added a clarifying part to page 8 paragraph 1 reading “These three genes were also found as significant MR evidences in our analysis. Integrating the MR results with the colocalization signals from the UKB-FinnGen meta-analysis and Nalls et al. signals as well as the pQTLs from the UKB-PPP and Nalls et al., we observe four proteins including GPNMB, BST1, CTSS, and HIP1R to be shared. Among which, GPNMB and BST1 have also been reported in CSF as significant MR signals by Kaiser et al. Given that our findings are based on plasma, such an overlap shows the great potential of these two proteins to be investigated as therapeutic targets.”

To clarify on UKB-FinnGen meta-analysis, we have added the following to paragraph 2 in page 5 reading: “Meta-analyzing UKB and FinnGen has added significant power to replicating many of known PD signals. The FinnGen GWAS signals itself shows two significant loci UBQLN4 and ARL17B in chromosomes 1 and 17, respectively. Upon meta analyzing it with the UKB GWAS, four other significant loci reached genome-wide significance including HIP1R, HLA-DRB1, GBA, and TMEM175. These established loci have been identified in the literature mainly in EUR population. Therefore, meta-analyzing UKB with FinnGen not only did not lead to spurious associations, but increased the number of loci reaching genome-wide significance. This can be attributed to similarities in the genetic architecture of the samples from the UKB and participants of FinnGen. We should note that all the samples in the UKB GWAS were selected from British ancestry”.

Reviewer 3

Author's Response: We would like to thank you for your deep and constructive comments which led to significant improvements to the manuscript.

Reviewer 4

I thank the authors for answering my questions, I have no further concerns with regards to publication.

Author's Response: We would like to thank you for your deep and constructive comments which led to significant improvements to the manuscript.

REVIEWERS' COMMENTS

Reviewer #2 (Remarks to the Author):

One of the key findings presented in the paper was for the potential causal role LGALS3. The authors now note that this was based on analysis “we are not confident about”, and have removed any mention of coloc. Does this now mean they no longer believe it to be a potentially causal factor? If so, does this impact their summary and conclusions on the relevance of this gene? I thought this was one of the key findings of the paper, but it now seems less clear.

Reviewer #3 (Remarks to the Author):

Responses to Reviewers' Comments

Summary of changes

We would like to thank the editor and the reviewers for your evaluation of our manuscript (NCOMMS-23-34002C) titled 'Proteogenomic network analysis reveals dysregulated mechanisms and potential mediators in Parkinson's disease'. We have addressed the comments raised by the Reviewer 2. Please find below our point-by-point responses to the reviewers' specific comments.

Reviewer 2

Reviewer's comment: One of the key findings presented in the paper was for the potential causal role LGALS3. The authors now note that this was based on analysis "we are not confident about", and have removed any mention of coloc. Does this now mean they no longer believe it to be a potentially causal factor? If so, does this impact their summary and conclusions on the relevance of this gene? I thought this was one of the key findings of the paper, but it now seems less clear.

Author's Response: Thanks for raising this point. In fact, this does not devalue the finding of LGALS3, as it is a significant MR hit, both in our results as well as the literature, and one of the top findings from our conducted network analyses. Therefore, it does not impact the abstract or conclusions that we have made around this gene and it is still one of the key findings of our manuscript. The sentence that we removed was to check if it is a coloc signal or not, which was not the case, which is also aligned with the literature. We hope this clarifies.